# Hierarchical Representational Transformations of Working Memory in Brains and Machines

## Abstract

Working memory (WM) maintains past inputs while processing new ones, yet how representations transform between encoding and retrieval remains unclear. Clarifying whether these representations are sustained through stable coding formats, dynamically updated subspaces, or their interplay is key to uncovering the mechanisms of WM. To address this, we combined high-resolution 7T fMRI from the Natural Scenes Dataset with recurrent neural networks (RNNs) trained on a naturalistic 1-back task. Using representational similarity, cross-decoding, and subspace geometry analyses, we directly compared rotational and non-rotational transformations between WM encoding and retrieval phases in brain regions and model layers. Our analyses revealed convergent evidence for a mixture mechanism of WM coding for encoding and retrieval information: early visual regions (V1–hV4) underwent large representational changes across encoding to retrieval phases, including both rotational and non-rotational transformations. Whereas higher-order regions in the prefrontal cortex (FEF, dlPFC) were more stable. Applying the same analyses to models showed a similar mechanism across layers, but critically depended on the learning objective and the recurrent architecture. We examined two different encoder architectures, ResNet and Vision Transformer (ViT), each trained with supervised and self-supervised learning objectives. Models with supervised encoders preserved a hierarchical layer dissociation paralleling the cortical gradient in both rotational and non-rotational transformations, while models with self-supervised encoders diverged in the rotational transformation. Among recurrent architectures, gated architectures (GRU, LSTM) better reproduced the brain-like mixture of subspace rotational transformation. Taken together, these results established hierarchical shifts between flexibility and stability in WM representational transformation in both humans and machines, with supervised learning objectives combined with gated recurrent dynamics most closely resembling human WM mechanisms.

## 1 Introduction

Working memory (WM) is the neural and cognitive process that temporarily stores and manipulates sensory information (Baddeley, 1992; 2003). It supports a wide range of higher-order cognitive functions such as learning, reasoning and decision-making (Collins & Frank, 2012; Daneman & Carpenter, 1980; Süß et al., 2002; Wagner, 1999; Engle, 2010; Cools & D'Esposito, 2011). Classic work has highlighted the limitations of WM in capacity and precision: its precision decreases with the number of items stored (Luck & Vogel, 1997; Ma et al., 2014) and with longer delays (Pertzov et al., 2017; Magnussen et al., 1998; Shin et al., 2017). Despite its limited capacity, WM exhibits a high degree of flexibility. People can maintain WM content even in the presence of new incoming stimuli. Such flexibility indicates that WM representations are not just mere traces of the original inputs, but undergo dynamic transformation to serve diverse cognitive demands.

The representational format of WM content remains an ongoing debate in neuroscience and cognitive science. WM-related signals observed in early visual cortex have been interpreted as evidence that WM and perception share a similar representational format (Serences, 2016; Christophel et al., 2017; Harrison & Tong, 2009). However, a shared format across different stages of information pro-

cessing could lead to interference between types of information that should be kept separate. Consistent with this notion, studies have shown that WM representations can differ from their perceptual counterparts. Such differences may naturally emerge during memory delays (Kwak & Curtis, 2022; Li & Curtis, 2023; Spaak et al., 2017; Murray et al., 2017), in the presence of sensory distractions (Xu, 2024; 2025; Libby & Buschman, 2021; Degutis et al., 2025), or when the WM content is assigned with different levels of priority (Wan et al., 2020; 2022). Reconciling these observations, some electrophysiological evidence suggests that a dynamic neural code and a stable neural code may coexist when maintaining WM information (Stokes et al., 2013; Murray et al., 2017).

In this study, we investigate how WM representations are transformed from encoding to retrieval, the stage where stored information is accessed to guide behaviors, as in a 1-back task requiring comparison between past items held in memory and the current input. Retrieval is particularly important because it is the stage when stored information is accessed and used for comparison, such as in an n-back task where past items must be retrieved to evaluate against the current input. We aim to characterize these transformations across brain regions along the cortical hierarchy. Specifically, we ask whether different stages of the cortical hierarchy implement distinct transformations: some regions may represent retrieved information in a format distinct from currently encoded signals, thereby reducing interference, whereas other regions may preserve a more coherent code between encoding and retrieval. By analyzing representational similarity and geometry, we aim to evaluate both rotational and non-rotational representational transformation. In addition, we compare the transformations from WM encoding to retrieval between human neural recordings and recurrent neural networks (RNNs) trained to perform continuous WM tasks. By testing neural networks with varying architectures and learning objectives, we aim to further identify the factors in models that support human-like representational transformation in neural networks.

To address this, we combine high-resolution 7T fMRI from the Natural Scenes Dataset (NSD) (Allen et al., 2022; Gifford et al., 2025) with RNNs trained on a 1-back WM task. We analyze representational similarity, cross-phase decoding, and geometric subspace rotation angles to test whether memory representations are best explained by stable or dynamically updated subspace mechanisms across the cortical hierarchical biological and artificial systems. Our contributions are as follows:

- **Mixture of dynamic and stable WM subspaces across the hierarchy**: Encoding–retrieval representations are partially overlapping but systematically transformed along the hierarchy, with greater transformation in early visual regions (e.g. V1-hV4), and more stable subspaces in higher-order regions (FEF, dlPFC) in both rotational and non-rotational transformation.
- **Model–brain alignment in WM encoding-retrieval transformation depends on both the encoder learning objectives and the recurrence modules**: Supervised models and gated architectures (GRU, LSTM) better captured the brain-like mixture of WM subspace transformation along the layers.

Together, these findings provide new insights into how the brain and machines using dynamic transformations to minimize interference in early regions while maintaining stable subspaces in higher-order regions to ensure reliability in WM. Furthermore, this efficient mixture of coding strategies is supported by the supervised learning objective and gated recurrent architectures in machines.

## 2 RELATED WORKS

RNNs are widely used in neuroscience to model WM due to their ability to maintain information over time and generate dynamics resembling cortical activity observed in humans and non-human primates (Wang, 1999; Wimmer et al., 2014; Compte et al., 2000; Bouchacourt & Buschman, 2019; Wang, 2021; Yang & Wang, 2020; Esnaola-Acebes et al., 2022). By training with WM delayed-response tasks, RNNs can reproduce hallmarks of WM phenomena such as persistent activity, attractor dynamics, and flexible subspace organization. These capabilities make RNNs a powerful computational framework for probing how WM representations are formed, maintained, and adapted under varying task demands.

Despite extensive research on neural representations of WM, studies linking representational transformations in neural networks and brains remain rare. Prior work has compared RNNs to post-cue WM tasks in non-human primates (Piwek et al., 2023) and human whole-brain EEG in n-back tasks

(Wan et al., 2022), but these studies relied on simple artificial stimuli (e.g. color and orientation) in predefined subspaces. As such, they could not capture representational dynamics for naturalistic stimuli, or dissociate coding strategies across brain regions given the low spatial resolution of EEG signals. Some studies analyzed representational transformations in human fMRI with simple artificial stimuli (Kwak & Curtis, 2022; Li & Curtis, 2023; Degutis et al., 2025) or high-dimensional naturalistic stimuli (Xu, 2024; 2025; Nakamura et al., 2025), but without comparing the neural representations of the brain to those of neural networks. More recently, transformation dynamics have been quantified in RNNs trained on n-back tasks with naturalistic stimuli (Lei et al., 2024), but without comparing RNNs with human brain data. These efforts face common limitations: reliance on simplified stimuli, scalar subspace metrics ill-suited to high-dimensional geometry, and isolated analyses of either neural networks or brain data. We address these gaps by directly comparing WM encoding-retrieval transformation matrices between artificial and biological systems, by using representational similarity, cross-decoding, and geometric alignment analyses.

## 3 METHODS

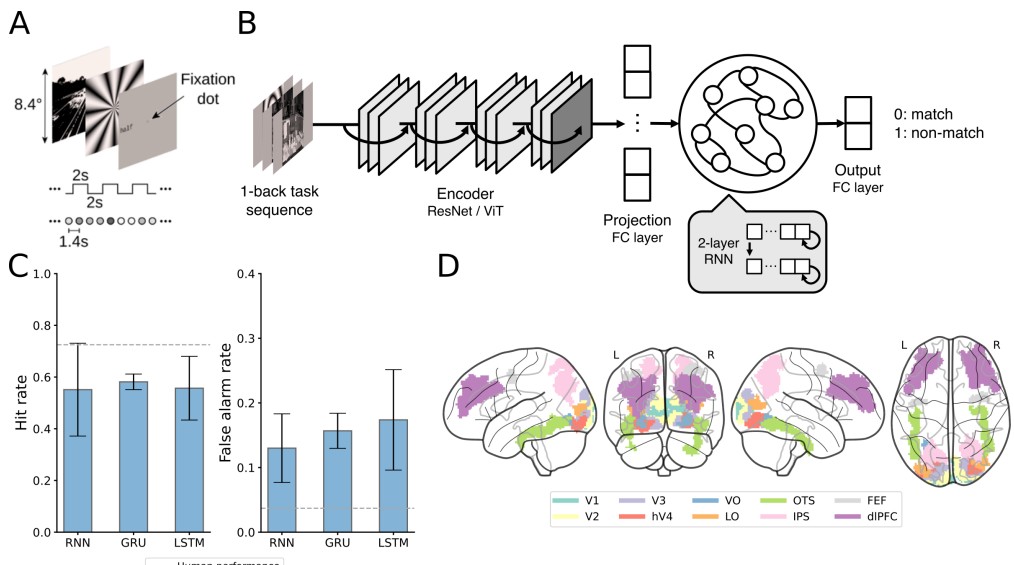

Figure 1: Tasks, model architecture, performance, and brain ROIs. (A) Trial design of the task. For 1-back task, subjects were asked to judge whether the currently presented image was identical to the previously one. Image from Gifford et al. (2025). (B) Model architecture. The model largely consists of these parts: encoder (ResNet, ViT) with different training objective (supervised, self-supervised learning), and RNN (vanilla RNN, LSTM, GRU). The input was the 1-back task stimuli sequences, and the encoder part extracted the representation, which was passed to a project layer in order to match the RNN's input dimension. Then RNN module was linked to a fully connected layer to produce the decision. (C) Model performances on the one-back task. Error bars show the mean ± 1 s.e.m across averaged testing performance over all the testing sets of models across 4 encoder types. (D) Regions of interest. Subject 1 from NSD was used as the sample subject for visualization.

### 3.1 NEURAL RECORDINGS AND TASKS

The NSD-synthetic dataset extends the Natural Scenes Dataset (NSD-core; Allen et al., 2022) with 7T fMRI recordings of fixation and 1-back tasks using new synthetic stimuli (Gifford et al., 2025). Data were collected from 8 participants viewing 284 synthetic images spanning diverse formats and semantic information, including 8 subclasses such as natural and manipulated scenes, noise, words, gratings, and chromatic variations.

Participants alternated between two tasks: a fixation task measuring perceptual representation and a 1-back task probing WM encoding and retrieval. Each task was performed in 4 runs. Each run

contained 93 trials, with each trial consisting of a 2-s image presentation followed by a 2-s inter-trial interval. In the fixation task, participants responded to color changes in the central dot, while in the 1-back task they indicated whether the current image matched with the stimulus before it (Fig. 1A).

The analyses were focused on 10 regions of interest (ROI, Fig. 1D), ranging from early visual areas (V1–hV4) to mid-level regions (LO, VO, IPS, OTS) and higher-level regions such as FEF and dlPFC, defined by the mask provided by the NSD (Allen et al., 2022), a probabilistic map (Wang et al., 2015), and an established parcellation atlas (Glasser et al., 2016). The more detailed description of the fMRI dataset and the tasks is in Appendix A.2.

## 3.2 MODEL

We modeled the 1-back task with a two-stage architecture (Fig. 1B). The frozen visual encoder encoded each image to a feature vector, which was passed through a fully connected project layer before being processed by a recurrent module. Note that the project layer and recurrent modules were trained in this work. The recurrent module consumed sequences of projected feature vectors, and emitted binary decisions, indicating whether the current stimuli matched the last one. For representational analyses, we extracted activations from the the project layer and recurrent hidden layers (layers 1 and 2). Implementation details are provided in Appendix A.3.

**The frozen encoder and projection module.** We used two different frozen encoder architectures: a convolutional neural network (ResNet-50; He et al., 2016) and a Vision Transformer (ViT-B/16; Dosovitskiy et al., 2020). We tested four frozen image encoder architecture–learning objective setups: ResNet-50 and ViT-B/16, each pretrained with ImageNet-1K supervised learning (SL) or self-supervised learning (SSL). We used contrastive learning objectives (He et al., 2020; Chen* et al., 2021) for self-supervised learning (see Appendix A.3 for details). These encoders remained frozen during training, and their final-layer feature outputs were processed by the project layer to reduce the dimension before the recurrent module. We extracted 4 encoder layers of each setup: one per block from ResNet-50's 4 blocks, and layers 3, 6, 9, 12 from ViT-B/16.

**Recurrent modules.** We evaluated 3 recurrent architectures—vanilla RNN (Elman, 1990), LSTM (Hochreiter & Schmidhuber, 1997), and GRU (Chung et al., 2014)—each with two layers. Our design crossed encoder types (ResNet vs. ViT), learning objectives (supervised vs. self-supervised), and recurrent modules (RNN, LSTM, GRU), enabling systematic comparison of architectures and learning objectives on human-like sequence processing. For brevity, we denote the 4 encoder–objective setups as ResNet-SL (ResNet-50, supervised), ResNet-SSL (ResNet-50, self-supervised), ViT-SL (ViT-B/16, supervised), and ViT-SSL (ViT-B/16, self-supervised).

## 4 RESULTS

### 4.1 MODEL PERFORMANCE

We trained 12 model configurations with 2 different learning objects $\times$ 2 encoder architectures $\times$ 3 recurrent architectures (details in A.3.5) with NSD-core images (Allen et al., 2022). Across all configurations, training accuracy exceeded 90%, and validation accuracy exceeded 85%. Next, these trained models were tested on the NSD-synthetic dataset (Gifford et al., 2025), which consists of 284 out-of-distribution (OOD) images. We trained the models with varying levels of "match" events, while the test set NSD-synthetic contained a highly imbalanced distribution of match events (11% targets). In the imbalanced environment, quantifying performance with hit rate and false alarm rate is more reliable than raw accuracy.

Here we reported the test performance with NSD-synthetic (Fig. 1C), where each of 12 models was trained with 5 random seeds (details in A.3.6). All recurrent architectures (vanilla RNN, GRU, LSTM) achieved moderate hit rates in the 1-back task, while lower than the human observers. Additionally, models exhibited higher false alarm rates than humans, suggesting a higher bias. Together, these results indicate that while models could perform the 1-back task, they do not fully reproduce the human decision strategy due to a mismatch between the match frequency of the training and testing environment.

## 4.2 NEURAL MECHANISM OF CONCURRENT WM ENCODING AND RETRIEVAL

After confirming that our models successfully learned the 1-back task, we next turned to our central question: How are representations transformed between encoding and retrieval? Specifically, we tested three competing assumptions about the mechanisms of processing concurrent WM encoding and retrieval information: (I) Stimuli held in WM maintain a stable representational format preserved across encoding and retrieval. (II) Representations are dynamically updated as the trials unfold, with systematic transformations between the encoding and retrieval phases. (III) A mixture of stable and transformed memory subspaces, and different cortical areas may implement different strategies.

To adjudicate between these accounts, we analyzed human fMRI and model activations using representational similarity, cross-decoding, and subspace geometry. These analyses revealed consistent evidence for the mixture mechanism, with early visual regions (V1–hV4) exhibiting stronger dynamic transformations and higher-order regions (dlPFC) maintaining more stable subspaces.

### 4.2.1 REPRESENTATIONAL SIMILARITY EVIDENCE

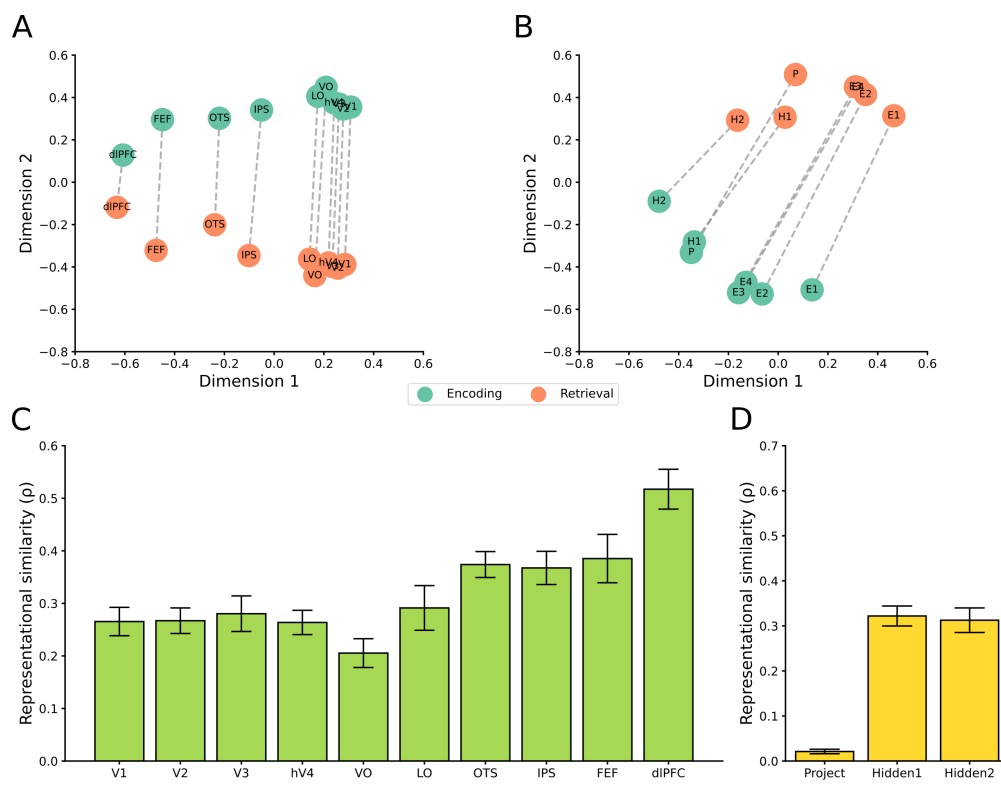

Figure 2: Representational similarity results. (A) MDS visual-path of the dissimilarity matrix based on RDMs of encoding and retrieval representations for each human ROI. (B) MDS visual-path of the averaged dissimilarity matrix across 12 models based on RDMs of encoding and retrieval representations for each model layer. E denotes frozen weight encoder layers, P denotes the project layer, and H denotes hidden layers. The number following the letter denotes the order of that layer. (C) Similarity scores (Spearman's $\rho$) of encoding and retrieval RDMs for each human ROI. Error bars represent ± 1 s.e.m across 8 subjects. (D) Similarity scores (Spearman's $\rho$) of encoding and retrieval RDMs for each model layer. Error bars represent ± 1 s.e.m across 12 models

With representational similarity analysis on brains and models, we consistently observed an increasing similarity scores between encoding and retrieval representations from low-level to high-level cortical regions or model layers. For the brain data, we constructed the identity-based representational structures (RDMs) for each ROI in both encoding and retrieval phases. We then calculated the dissimilarity (1 - Spearman's $\rho$) of the RDMs from both phases and all ROIs. Finally, we ap-

plied multidimensional scaling (MDS) to project these relationships into a two-dimensional space (Fig. 2A, with methodology details in Appendix A.7.1).

Within each phase, the representations of low-level (V1-hV4) and mid-level areas (LO and VO) clustered close to each other, while the representations shifted a lot from low-level to the higher-level cortical areas (dlPFC). Interestingly, the WM encoding representations in the lower areas (V1-hV4) were very distinct from their representations in the retrieval phase, while the representations in the higher-level areas (dlPFC) exhibited larger similarity across the phases. This similarity between the encoding to retrieval phase increased gradually along the hierarchy. RNNs exhibited the same hierarchical pattern, as the representations in two phases were closer in the higher-level hidden layers compared to the lower untrained encoder layers and trained project layer (Fig. 2B; Fig. A6 for model-wise visualizations). In the following analyses, we mainly focused on the project and 2 hidden layers trained on the 1-back WM task.

We further confirmed our observation in the MDS of RDM-dissimilarity by directly computing the similarity score Spearman's $\rho$ for each brain ROI or model layer between RDM of encoding v.s. retrieval phase (Fig. 2C). A permutation based one-way ANOVA test showed a significant main effect of ROI on the similarity score ($p < .001$), where the score was highest in dlPFC compared to all the other regions (permutation post-hoc test, $p < .001$, FDR corrected). For the models, a one-way ANOVA revealed a significant main effect of layer on encoding–retrieval similarity ($p < .001$). FDR post-hoc paired t-tests showed higher similarity for both hidden layers than for the project layer ($p < .001$, Fig. 2D). To assess the reliability of the RSA, we shuffled RDM cells and ran the same set of analyses (Fig. A7). A permutation based repeated-measures ANOVA revealed significant main effects of observed or shuffled condition and ROI ($p < .001$), as well as a significant interaction between condition and ROI ($p < .001$). FDR post-hoc paired t-test between the observed pattern and shuffled conditions (random baseline) confirmed the robust effects across all ROIs ($p < .001$).

Overall, the evidence of relative representational distance along stimuli suggested hierarchical changes from dynamically updated memory subspaces in lower regions/layers to more stable representational formats in high-level regions/layers, in both biological and artificial organizations.

### 4.2.2 DECODING EVIDENCE

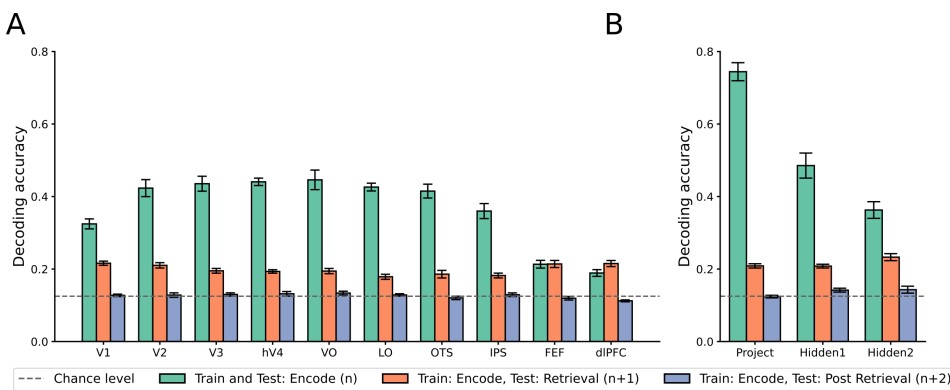

Figure 3: Decoding accuracies of the decoder trained in the encoding (n) phase, tested in the encoding (n), retrieval (n+1), and post-retrieval (n+2) phases. (A) Results of each human ROI. Error bars represent ± 1 s.e.m across subjects. (B) Results of each model's layer. Error bars represent ± 1 s.e.m across models.

To complement RSA, we next used cross-decoding to investigate representational changes between the encoding and retrieval phases (Harrison & Tong, 2009; Xu, 2025). For each ROI or model layer, we trained linear decoders to classify the stimulus identity across 8 subclasses (Gifford et al., 2025). Decoders were trained during the encoding phase with the current stimuli (step $n$) and evaluated either within the same phase ($n$), during the retrieval trial ($n+1$) when a match/no-match action is required, or during the post-retrieval trial ($n+2$) when the stimuli information is irrelevant (details in Appendix A.7.2).

As shown in Fig. 3A, decoding accuracies were highest when trained and tested within encoding, of which all ROIs showed above chance accuracy (permutation test, $p < .01$). The decoder trained in the WM encoding phase can reliably generalize to the retrieval ($n+1$) phase in all ROIs, achieving above chance level decoding accuracy (permutation test, $p < .01$). However, the decoder trained in WM encoding phase tested in post-retrieval ($n+2$) phase was not above chance level in all ROIs (permutation test, $p > .05$). Therefore, the decoders in frontal areas are reliably generalized from encoding to retrieval phase, but not the post-retrieval phase when the encoded information is irrelevant. More importantly, there is a significant 2-way interaction of ROI and test condition ($p < .001$): the decoders of most the ROIs showed higher decoding accuracies when tested in encoding phase compared to the retrieval phase (FDR corrected, permutation based post-hoc paired t-test, $p < .001$), except for frontal areas (FEF and dlPFC $p > .05$). This significant cross-decoding performance drop could signal a large representational transformation (Xu, 2025) between encoding and retrieval phase in the low- to mid-level visual regions (V1-IPS). Additionally, all ROIs showed above chance accuracy when the decoders were trained and tested in the retrieval (n+1) phase (Fig. A1, permutation test, $p < .01$), indicating an active engagement of the all ROIs, including FEF and dlPFC, during the memory retrieval.

Decoding accuracy in the fixation task for decoders trained and tested in the encoding phase didn't achieve above chance accuracy across all ROIs, and lower than that in the 1-back task (permutation test, $p < .01$), indicating that 1-back decoding reflected WM processes beyond sensory responses.

For RNNs, decoding showed a pattern similar to humans, as shown in Fig. 3B. The decoder trained in encoding could reliably generalize to the retrieval but not post-retrieval phase. A 2-way ANOVA showed an interaction between layer and test condition ($p < .001$) and a main effect of layer and test condition ($p < .001$). For decoder train and test in encoding phase, the decoding accuracy was highest in the project layer, and then the 1st hidden layer, with lowest accuracy in the 2nd hidden layer (FDR post-hoc paired t-test $p < .001$). When tested in the retrieval phase, the accuracy was higher in the 2nd hidden layer than the project layer (FDR post-hoc paired t-test $p < .05$).

Hence, consistent with RSA results, decoding results also suggest that stimulus information is well-represented at encoding but undergoes substantial transformation from encoding to retrieval. Importantly, RSA is insensitive to subspace rotations since it is built from pairwise representational distances (Fig. 4D). This motivates our geometric rotation analyses, where we directly quantify the rotation angles between encoding and retrieval subspaces to test for stable versus dynamic subspace mechanisms.

### 4.2.3 GEOMETRY ROTATION EVIDENCE

To directly assess the rotational geometric relationship between the encoding and retrieval representations, we computed the principal angles between the subspaces spanned by class-selective decoder weight vectors in each ROI (Fig. 4A) (Gower, 1975). We first used the top-1 principal angle to quantify the degree of rotations of the neural subspaces between the encoding and retrieval phases. This top-1 angle represents the neural dimension that is most aligned across the two phases. We observed a significant main effect of ROIs in the top-1 angle (permutation test, $p < .05$), and there was a reduction in rotations when moving from low-level visual areas to higher-level brain regions (FEF, dlPFC). We conducted the same analysis using the average of the top-2 angles, which is a more robust measure of rotation, and found similar results: there was a significant main effect of ROIs (permutation test, $p < .01$), and the rotation angle between encoding and retrieval was larger in the low-level regions (V2, V3) compared to the high-level frontal areas (FEF, dlPFC; post-hoc paired t-tests, FDR corrected $p < .05$). The permutation based 2-way ANOVA (2 angles x 10 ROIs) showed no significant interactions ($p > .05$). Therefore the pattern of the rotation angles across ROIs remains across the angle dimensions.

As a validation of whether decision geometry across the WM phases can be approximated by a rigid rotation, for each ROI we trained a classifier in the encoding subspace and tested it on retrieval with vs. without applying the rotation derived from principal angles (details see Appendix A.7.2). Across ROIs, the rotation improved cross-phase decoding when we kept the most aligned Top 1–2 axes (one-sample t-test $p < .05$). In other words, only smaller portion of well-aligned dimension of subspaces carry transferable signals with rotation, whereas full-dimensional representational transformations are explained by non-rotational reconfigurations (such as the relative representational distance).

Combined with the RSA results, these findings indicate that, while encoding and retrieval share some stable representational subspaces in higher-order regions, the representational transformation in the low- to mid-level regions undergoes systematic transformation, in both rotational and non-rotational manners. Together, these findings indicate a WM coding strategy that leverages dynamic transformations in lower-level regions to minimize interference, while maintaining stable representations in higher-order regions to preserve reliability.

### 4.2.4 The effect of task objectives and recurrent architectures in rotation

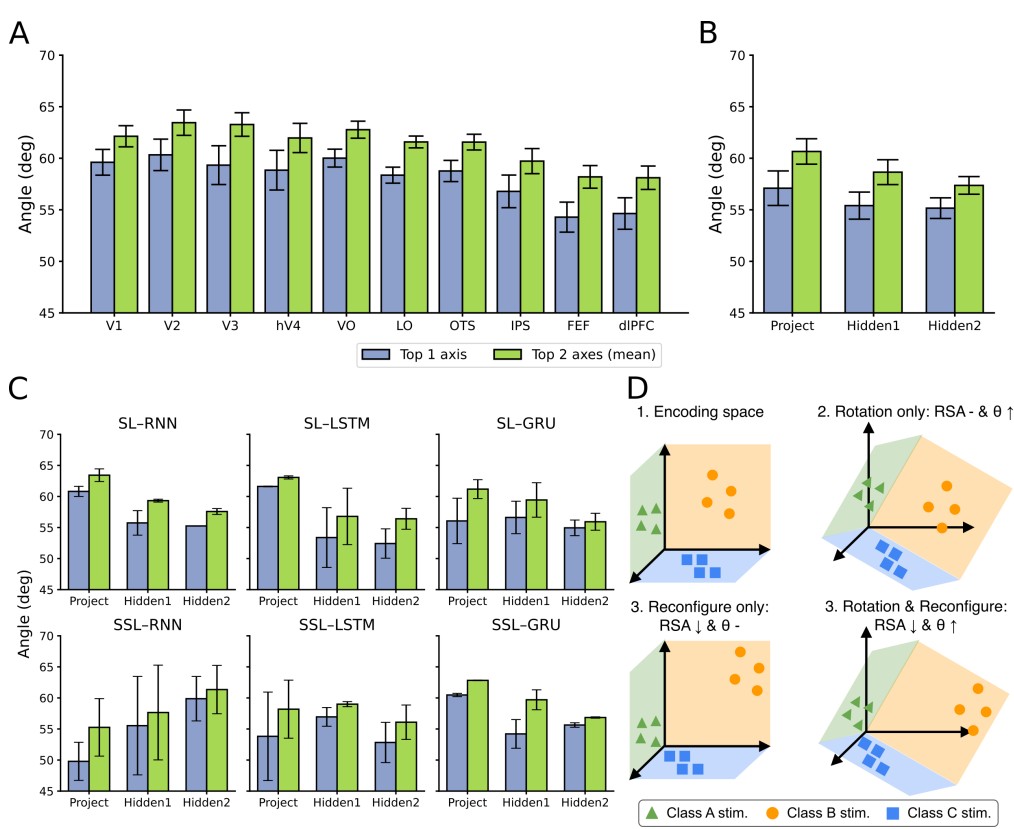

Figure 4: Geometry analysis results. (A) Rotation angles of the top-1 axis and top-2 axes (mean) from encoding to retrieval in each ROI. Error bars represent ± 1 s.e.m across 8 subjects.(B) Rotation angles of the top-1 axis and top-2 axes (mean) from encoding to retrieval in each layer of the models. Error bars represent ± 1 s.e.m across 12 models. (C) Group average rotation angles of the top-1 and top-2 axes (mean) from encoding to retrieval in each layer of the models. SL denotes supervised, SSL denotes self-supervised. We averaged the rotation angles of Resnet-50 and Vit-B/16, since the patterns were consistent across encoder architectures. Error bar represent ± 1 s.e.m across models with varying encoder architectures.(D) Example illustrations of the representation transformation. The example representation space in the encoding phase (1), and how rotational or/and non-rotational transformation from encoding space to retrieval space would influence RSA and rotation angle $\theta$ (2-4). - denotes no difference, ↑ denotes increase, ↓ denotes decrease.

To have a direct comparison between the human brains and artificial systems, we next applied the same subspace rotation analysis on the RNNs. Interestingly, the models also exhibited smaller rotation angles (greater subspace alignment) in the higher hidden layer compared to the lower layers (Fig. 4B) similar to the brain data. The ANOVA revealed a significant main effect of the layer on the rotation angle ($p < .001$), without an interaction between layer and the angle types ($p > .05$). However, there was no significant difference between layers in post-doc paired t-tests ($p < .05$), suggesting the hierarchical pattern of rotation in model might not as consistent as humans.

We further investigate the factors under the divergence of the rotation angles in human and machines by separately looking into the results of models with different learning objectives of the encoder (Supervised learning -SL vs Self-Supervised learning -SSL), encoder architectures (ResNet vs ViT), and recurrent architectures (vanilla RNN, GRU and LSTM) (Fig. 4C).

**The effect of learning objectives.** Overall, models that used supervised learning (SL) encoder showed shared mechanism with humans than those that used self-supervised learning (SSL): For the SL models, the project layer showed lager rotational angles which parallel with early to mid-level ROIs (V1–hV4, VO-IPS), while the hidden layer exhibited smaller rotations similar to the higher-order regions (FEF, dlPFC; upper row of Fig. 4C compared to the lower row). This layer-dependent shift parallels the hierarchical gradient observed in the brain (Fig. 4A). By contrast, self-supervised models did not display such gradient across layers: The rotation angle in the project layer was not consistently higher than the hidden layers as in humans, suggesting that self-supervised pretraining induces representational rotations that diverge more strongly from the transformations measured in cortex.

**The effect of architectures.** Among the recurrent architectures, we also observed systematic but distinct rotation patterns. GRU and LSTM exhibited the layer-dependent shift (Fig. 4C columns 2-3): its project layer showed larger rotation angles which is similar to human early and mid-level visual ROIs (V1–hV4, VO, LO, OTS), while hidden layers showed smaller rotational angles mapping comparably onto higher-order regions (FEF, dlPFC). By contrast, vanilla RNN models (especially the self supervised ones) exhibited a different profile: the project layer showed smaller rotation angles compared to the hidden layers. Additionally, we didn't observe the effect of the encoder architecture (ResNet vs ViT) in the rotational transformation between WM phases.

## 5 DISCUSSION

**Hierarchical mixture of stable and dynamically updated memory subspaces.** Our findings support the view that WM representations combine both stable and dynamic components. Lower-level visual areas tend to reformat representations across phases, whereas higher-order regions such as prefrontal cortex exhibit more stable coding. The co-existence of dynamic and stable neural code echoes the electrophysiological studies suggesting monkey's prefrontal cortex could adapt to behavioral demands (Stokes et al., 2013; Murray et al., 2017), whereas the current work consider various human ROIs, which parallels the layers in neural networks. The increasing stability of WM representations along the cortical hierarchy is consistent with recent fMRI work on WM representations using simpler stimuli (Li & Curtis, 2023). Additionally, merging evidence in comparing perception to visual imaginary or long-term memory have also shown a stable representations across phases in high-level cortical areas (Favila et al., 2022; Breedlove et al., 2020). Together, these findings suggest that stable coding may emerge prominently in higher-order cortex for higher reliability, whereas low-level areas undergo larger transformations for lower interference.

**The impact learning objectives in rotational WM transformation.** By training artificial neural networks on the same WM tasks performed by humans, we observed greater rotational transformation of encoding–retrieval representations in the early project layer than in the late hidden layers, paralleling the results in human brains. However, this shared mechanism between human and machine strongly depended on the learning objectives of the encoders. Supervised encoders preserved a hierarchical project–hidden dissociation that paralleled the cortical gradient, whereas self-supervised encoders diverged more strongly from this hierarchical structure. This suggests that supervised training enforces categorical abstraction across layers, which in turn constrains how memory subspaces are rotated across time. However, self-supervised contrastive objectives prioritize invariance over categorical structure, which may distort the temporal transformations relevant for WM (Konkle & Alvarez, 2022).

Pervious studies in brain-model alignment have centered on assessing models based on its brain activation predictability, and reported mixed results comparing SL and SSL models. Several studies have found advantages of certain SSL models (Zhuang et al., 2021), while others highlights similarities or even SL advantage (Conwell et al., 2024; Konkle & Alvarez, 2022; Khaligh-Razavi & Kriegeskorte, 2014). The current study did not evaluate models based on the brain predictability, whereas we investigated whether models exhibit shared transformation across WM phases, leading to a different alignment criterion in our work. On the other hand, when we consider the emer-

gence of human-like representational transformation mechanism as a subset of the above topic, our conclusion aligns with previous studies showing SL advantage.

**Gated recurrent mechanism supports human-like WM transformation.** Across different types of recurrent modules, GRUs and LSTMs showed decreasing rotational transformation from WM encoding to retrieval phases across layers, similar to humans. In contrast, vanilla RNNs exhibited inconsistent patterns to humans, suggesting that insufficient control over information flow may fail to capture the balance of stability and dynamics observed in the cortex during WM. Overall, these results indicate that gated recurrent dynamics in GRUs and LSTMs are essential for modeling the coexistence of stable and dynamic subspaces in the brain.

**Out-of-distribution generalization.** In the current study, the models were trained and validated on a subset of the NSD-core, while tested on NSD-synthetic, an out-of-distribution (OOD) dataset. Model performance was lower than on in-distribution NSD-core, highlighting a distribution shift. This likely reflects two potential limitations: (I) the frozen encoder may not fully capture task-relevant features for novel stimuli, and (II) recurrent backbones tend to maintain features learned during training, which can be hard to adapt to unseen inputs. In contrast, humans readily generalize WM across diverse visual inputs, reflecting the brain's ability to flexibly abstract stimulus features while preserving task-relevant geometry. Prior work shows that human vision achieves robust OOD generalization by leveraging hierarchical abstractions and flexible context adaptation (Fang & Sims, 2025). These comparisons underscore that while models can approximate in-distribution representational dynamics, OOD generalization remains a challenge for both supervised and self-supervised vision models (Geirhos et al., 2021). Future work should explore adaptive or hierarchical encoders that can dynamically preserve task-relevant features across domains.

## 6 LIMITATIONS

The current work focused on the 1-back task, which is widely used to investigate the WM neural mechanisms, combined with human fMRI (Malisza et al., 2005; Ricciardi et al., 2006; Lee et al., 2013; Ateş et al., 2017) and EEG (Audrain et al., 2020; Gjini et al., 2007). Including multiple WM tasks would further strengthen the generalization of our findings in WM transformation across encoding to retrieval phases. However, NSD-synthetic 1-back dataset (Gifford et al., 2025) is currently the only available WM benchmark that provides high-resolution, single-trial fMRI data on naturalistic stimuli. Because the focus of the present study is on shared mechanisms between models and humans, we are therefore constrained by the availability of such biological ground truth. To partially address this limitation on the modeling side, we analyzed models with fine-tuned encoders trained on the 1-back task (Fig. A5) and additionally trained new models on a 2-back task (Fig. A4). The results converged to the same central conclusion: hierarchical representational transformations remain consistent regardless of task difficulty or whether encoders are frozen or fine-tuned.

The NSD-core (Allen et al., 2022) contains over 73,000 images, we sampled only 600 to build the 1-back training sequences. This choice kept training computationally feasible across multiple architecture–objective combinations. The subset spans all 80 categories, ensuring diversity while reducing training time. While more samples might improve performance, we expect relative comparisons (e.g., gated vs. non-gated recurrent modules, supervised vs. self-supervised encoders) to hold since all models used the same dataset, while larger training and testing datasets would strength our conclusions. Additionally, though models perform the 1-back task well, their error patterns differed from humans. Several models showed a higher bias toward predicting "match" compared to human participants. Thus, while the models captured aspects of WM representational transformation across phases, their ability to regulate decisions was weaker than humans. Though the bias result wasn't link to the model-brain alignment in the WM representation transformation, future work would benefit from comparing computational principles between humans and models exhibit more human-like behavioral patterns.

## ETHICS AND REPRODUCIBILITY STATEMENT

**Ethics statement.** This work uses a publicly available fMRI dataset released by Allen et al. (2022) and Gifford et al. (2025). As reported in the original dataset publication, informed written consent was obtained from all participants, and the experimental protocol was approved by the University of Minnesota Institutional Review Board. Our study did not involve any new data collection with human subjects. We adhere to the ICLR Code of Ethics in conducting and presenting this research.

**Reproducibility statement.** We provide detailed descriptions of the data preparation procedures and model training settings to ensure transparency and reproducibility. In addition, we plan to publicly release the processed data and the code used for analyses for reproducibility.

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

# A APPENDIX

## A.1 LLM USAGE DISCLOSURE

Large language models (LLMs) were employed solely as a tool to refine wording, improve clarity, and enhance the readability of the manuscript. LLMs were not used to generate original ideas, analyses, or conclusions. All substantive content, arguments, and interpretations presented in this work are from authors' own.

## A.2 NEURAL RECORDINGS AND TASKS

The NSD-synthetic dataset extends the Natural Scenes Dataset (NSD-core) with high-resolution 7T fMRI recordings, providing single-trial beta estimates across fixation and 1-back tasks with new synthetic stimuli (Gifford et al., 2025).

**Participants and sequences.** The NSD-synthetic dataset consisted of data from eight subjects who had completed the NSD-core experiment (Gifford et al., 2025) and subsequently participated in the NSD-synthetic experiment, in which 284 synthetic images were tested. The whole-brain BOLD was acquired at 1.8 mm isotropic resolution (TR = 1.6 s) using gradient-echo EPI sequences on a Siemens 7 T scanner.

**Stimuli.** 284 synthetic images were generated to vary in visual format and semantic content, including 8 subclasses - noise, natural scenes, manipulated scenes, contrast modulation, phase-coherence modulation, words, spiral gratings, chromatic noise (Gifford et al., 2025). The manipulated scenes are the transformed versions of the natural scenes, such as upside-down, Mooney and lie-drawing scenes. This design enabled systematic manipulation of stimulus identity and format.

**Tasks.** Two tasks were performed in alternating runs, consisting of 4 runs (93 trials per run). Each trial started with the presentation of an image for 2 s, followed by a 2-s inter-trial interval. (1) In the Fixation task, subjects detected color changes in the fixation dot and immediately pressed button, providing a baseline measure of pure perceptual representation. (2) 1-back WM task, in which subjects pressed a button to indicate whether the current image matched the immediately preceding image in identity, thus dissociating the encoding and retrieval stages.

**Preprocessing and GLM.** We used the preprocessed single-trial GLM beta estimates provided by NSD-synthetic (1.8mm with TR = 1.33s). The beta weights were estimated using GLMsingle—employing optimized HRFs, GLMdenoise, and ridge-regularized regression—to yield percent signal change estimates for each image under both task conditions (Prince et al., 2022).

**Regions of interest.** We included 10 regions of interest (ROIs) in various regions, from retinotopic visual regions to higher-order cognition regions. For V1, V2, V3 (grouping ventral and dorsal subregions for each corresponding ROI) and hV4, we used regions provided from NSD, which were manually drawn for each subject based on pRF experiment (Allen et al., 2022). Lateral occipital (LO; grouping LO1 and LO2), ventral occipital (VO; grouping VO1 and VO2) and intraparietal sulcus (IPS; grouping IPS0 to IPS5) were defined using maximum probability map (Wang et al., 2015). Occipito-temporal sulcus (OTS) was defined using "corticalsulc" ROIs provided by NSD (Allen et al., 2022). Frontal eye field (FEF) and dorsolateral prefrontal cortex (dlPFC) were defined using the HCP-MMP1.0 atlas by Glasser et al. (2016). To be specific, the dlPFC was defined as a grouped region of Brodmann areas 9, 46 and 9/46 (Petrides, 2005).

## A.3 1-BACK TASK MODELS DETAILS

In this section, we provide additional details to ensure full reproducibility of our experiments. We first expand on the design choices for the encoder and recurrent modules, followed by dataset construction, preprocessing, and training hyperparameters. Unless otherwise noted, all models were implemented in PyTorch (Paszke et al., 2019).

### A.3.1 ENCODERS

**ResNet-50.** We used the ResNet-50 implementation (He et al., 2016) from `torchvision.models`, with pretrained weights from two sources. (1) Supervised weights

trained on ImageNet-1K (`weights=ResNet50_Weights.IMAGENET1K_V2`) were down-loaded from PyTorch Hub[1]. (2) Self-supervised weights were obtained from MoCo v2 (He et al., 2020), released in the official GitHub repository[2], where we selected the 800-epoch pretrained checkpoint. For ResNet-50, features were extracted from the output of the last residual block (`layer4.2.relu_2`), which yields a 2048-channel representation.

**ViT-B/16.** We used the ViT-B/16 implementation (Dosovitskiy et al., 2020) from `torchvision.models`, again with pretrained weights from two sources. (1) Supervised weights trained on ImageNet-1K were provided in PyTorch Hub[3]. (2) Self-supervised weights were obtained from MoCo v3 (Chen* et al., 2021), released in the official GitHub repository[4], where we used the ViT checkpoint pretrained for 300 epochs. For ViT, features were extracted from the output of the final (12th) transformer encoder block (`encoder.layers.encoder_layer_11.add_1`), which yields a 768-dimensional representation.

### A.3.2 RECURRENT MODULES

**Vanilla RNN.** We implemented a two-layer vanilla RNN (Elman, 1990) with input size 512 and hidden size 256, using PyTorch's `nn.RNN`. The hidden state at each time step $h_t$ was updated by combining the projected encoder feature $f_t$ and the previous hidden state $h_{t-1}$ through a linear transformation followed by $\tanh$ activation. Dropout with probability $p = 0.3$ was applied to the hidden states of each layer. Parameters were initialized using Kaiming uniform initialization (He et al., 2015). At each time step, the hidden state from the last RNN layer was passed through a fully connected layer and sigmoid activation to yield a binary prediction $\hat{y}_t$.

**LSTM.** The long short-term memory (LSTM) (Hochreiter & Schmidhuber, 1997) was imple-mented with two layers and hidden size 256 with input size 512, using PyTorch's `nn.LSTM`. We adopted the standard gating mechanism (input, forget, and output gates) as in PyTorch's `nn.LSTM` implementation. The same dropout ($p = 0.3$) and initialization scheme as the vanilla RNN were used.

**GRU.** The gated recurrent unit (GRU) (Chung et al., 2014) was similarly implemented with two layers and hidden size 256 with input size 512, using PyTorch's `nn.GRU`. The GRU simplifies the LSTM by merging input and forget gates into a single update gate, thereby reducing parameter count. As with the other recurrent models, dropout was applied to hidden states and predictions were generated via a fully connected layer followed by sigmoid activation.

### A.3.3 ENCODER–RNN CONNECTION

To bridge the encoder and recurrent modules, we extracted image features from the final representa-tional layer of each encoder (ResNet-50: `layer4.2.relu_2`; ViT-B/16: `encoder_layer_11`). These features were high-dimensional (2048 channels for ResNet, 768 for ViT) and were adapted for recurrent processing through a projection module. Specifically, we first applied a pointwise ($1 \times 1$) convolution to reduce the feature dimensionality to match the RNN input size. The resulting features were then passed through a fully connected layer and subsequently normalized with layer normalization (Ba et al., 2016). We denote this processed representation as the *projected feature layer* (project layer), which served as the input to the recurrent module.

### A.3.4 DATASET GENERALIZATION

Images were sampled from the NSD-core dataset (Allen et al., 2022) and preprocessed following NSD-synthetic conventions (Gifford et al., 2025). Each image was center-cropped to a square using the smallest dimension and resized to $224 \times 224$ pixels. Synthetic images were incorporated by

---

[1] `https://docs.pytorch.org/vision/main/models/generated/torchvision.models.resnet50.html#torchvision.models.resnet50`

[2] `https://github.com/facebookresearch/moco`

[3] `https://docs.pytorch.org/vision/main/models/generated/torchvision.models.vit_b_16.html#torchvision.models.ViT_B_16_Weights`

[4] `https://github.com/facebookresearch/moco-v3`

concatenating the official NSD-synthetic with an additional collection of chromatic noise images. Invalid indices (e.g., corrupted crops) were removed prior to concatenation.

Training and validation sequences were generated using a 1-back trial generator. Each run consisted of 93 image presentations, with repeat events ($y_t = 1$ if $x_t = x_{t-1}$, otherwise $y_t = 0$) introduced at predefined probabilities $\{0.02, 0.1, 0.3, 0.5, 0.7\}$. The training set contained 40 runs (8 per probability condition), while the validation set contained 10 runs. During the data generation, we set random seed as 42.

rapped in PyTorch `Dataset` objects and loaded with the `DataLoader` API (batch size 1, shuffled). Each batch yielded sequences of images and labels suitable for recurrent training.

### A.3.5  MODEL TRAINING

To evaluate generalization, we trained and validated models on a subset of the Natural Scenes Dataset (NSD-core) (Allen et al., 2022), ensuring no overlap with the NSD-synthetic test set. From the 73,000 available images, we randomly sampled 600 spanning all 80 categories, which were used to generate 1-back training sequences. The training set comprised 40 runs and the validation set 10 runs, each with 93 image presentations. Repeat events (label = 1, when $x_t = x_{t-1}$) were inserted at controlled probabilities. All images were preprocessed following NSD-synthetic conventions (Gifford et al., 2025): center-cropped to a square using the shortest dimension and resized to $224 \times 224$ pixels.

Model parameters were optimized using Adam (Kingma & Ba, 2014). We used an initial learning rate of $1 \times 10^{-3}$ for GRU and LSTM parameters, $5 \times 10^{-4}$ for vanilla RNN parameters, and $2 \times 10^{-3}$ for the projection and classifier layers. A weight decay of $1 \times 10^{-4}$ was applied to all trainable parameters. Encoders remained frozen with pretrained weights, while recurrent modules were initialized with Kaiming initialization (He et al., 2015). Dropout ($p = 0.3$) was applied to RNN hidden states, the pointwise convolution, and the fully connected layers. All models were optimized with binary cross-entropy loss. Training was performed with batch size 1 on a single NVIDIA A100 GPU. We trained all the models for 90 epochs, and models with the best validation accuracy were selected for the following analyses of our work.

### A.3.6  RANDOM SEED TESTING

To ensure the stability of our 1-back models, we trained all architectures under multiple random initializations. Specifically, each model was trained using a distinct random seed (random seed set to 100, 200, 300, 400, 500) that controlled weight initialization, data shuffling, and three different RNN types state initialization. This procedure allowed us to assess whether the learned working-memory representations were consistent across training replicas.

For evaluation, we generated a new random test dataset using NSD synthetic images and a fixed master seed (random seed 2025). All trained replicas were tested on this identical test set, ensuring that performance variability reflected differences in the learned representations rather than stochasticity in test sequence construction. The test set followed the same structure as in the main 1-back protocol: 4 runs of continuous visual streams (sequence length $T = 93$), matched in stimulus composition across all model variants. We report both the mean and standard error of hit rate and false alarm rate across seeds for each architecture.

### A.4  2-BACK TASK MODELS

To evaluate whether our architecture supports higher-order working memory demands, we extended the framework to a 2-back variant of the task. In this paradigm, the model must determine whether the current stimulus $x_t$ matches the stimulus presented two steps earlier, $x_{t-2}$. The overall training and testing procedures closely mirror those used in the main 1-back experiment, with modifications described below.

### A.5  MODEL TRAINING

We trained hybrid visual–temporal architectures that combine frozen visual encoders (ResNet-50, MoCo v2, ViT-B/32, and MoCo v3) with a GRU-based recurrent head. Unlike the main 1-back

setup—where training was performed solely on NSD-core images—the 2-back extension employed a *mixed N-back training curriculum*. Specifically, each training batch interleaved 1-back and 2-back trials, requiring the GRU to flexibly maintain and update memory states across variable temporal horizons.

Training used the AdamW optimizer with a cosine-annealing learning-rate schedule. A masked cross-entropy loss was applied to accommodate variable-length label sequences generated by the mixed-N curriculum. Consistent with the original human behavioral experiment, all models were trained with the same target-to-true ratio (0.02, 0.1, 0.3, 0.5, 0.7).

### A.6 MODEL TESTING

Testing procedures were designed to be directly comparable across all architectures. As in the main 1-back experiment, we used NSD synthetic images as the evaluation stimuli. However, because no dedicated 2-back test split exists, we generated test sequences using fixed random seeds while maintaining the same evaluation structure used in the 1-back experiments.

**Protocol.** For each model, we conducted 10 independent test runs, each initialized with a distinct master seed (1000–1009). Every run consisted of three blocks of continuous visual streams (sequence length $T = 93$), ensuring robustness to random variation in trial composition.

**Stimuli.** Stimuli were constructed from NSD synthetic images following the same procedure as in the 1-back evaluation, but paired with 2-back labels.

**Metrics.** For each model, we computed task accuracy and extracted activations from all layers and task phases. These activations were used to construct representational dissimilarity matrices (RDMs) for later representational-geometry analyses.

#### A.6.1 2-BACK TASK PERFORMANCE

Across all four architectures, models achieved over 95% training accuracy and roughly 75% validation accuracy on the mixed 1-back/2-back training curriculum. When evaluated on randomly generated NSD-synthetic test sequences. For testing, all model performance accuracy higher than 65% accuracy.

### A.7 ANALYSIS

#### A.7.1 REPRESENTATIONAL SIMILARITY ANALYSIS

**Representational dissimilarity matrices (RDMs).** RDMs enable the comparison of representation across systems. The dissimilarity between two representations $X_i$ and $X_j$ can be expressed as $D(X_i, X_j) = 1 - \frac{\text{cov}(X_i, X_j)}{\sigma(X_i)\sigma(X_j)}$ , where $cov(X_i, X_j)$ denotes the covariance between the two representations, with $\sigma(X_i)$ being the standard deviation of $X_i$. This formula denotes a $1 - Pearson$ distance between the representations of stimuli $i$ and $j$.

The RDM is a symmetric $n \times n$ matrix $R$ where $R_{ij}$ reflects the dissimilarity between the representations of stimuli, resulting in a 284 by 284 matrix in the current study. Mathematically, the RDM is given by $R_{ij} = D(X_i, X_j) \quad \forall i, j \in \{1, 2, \ldots, n\}$ .

We constructed RDMs for each subject and each ROI in both encoding and retrieval phases of the fMRI recordings. Similarly, we constructed RDMs for each model and each layer in two phases based on the layer activations.

**Representational Similarity Analysis (RSA).** RSA provides a common framework to quantitatively compare relative representational distances as in RDMs across different modalities and phases of representations (Kriegeskorte et al., 2008). To assess the similarity of neural and model representations, we calculate the similarity score (Spearman's $\rho$) between the RDMs: $\rho = \text{Spearman}(\text{vec}(R_A), \text{vec}(R_B))$ .

We used Spearman correlation, between the encoding and the retrieval phase, of each brain ROI or model layer, to investigate how the relative representational distances across stimuli change from the encoding to the retrieval phase. Furthermore, we analyzed a path-based RSA to inspect the representational similarity changes along the visual hierarchy or model layers. Specifically, we treated the set of ROI-level or layer-level RDMs from each phase (encoding or retrieval) as a trajectory in representational space. RDMs were vectorized, and dissimilarities of RDMs across all ROIs (or layers) were computed as $1 - \rho$. The resulting dissimilarity matrix was then embedded in two dimensions using multidimensional scaling (MDS), allowing us to visualize each phase as a continuous path through representational geometry.

**Shuffling RSA for reliability.** To evaluate the reliability of the RSA, we performed the same set of RSA with randomly shuffling the rows and columns of the RDMs within each subject. For each ROI, permuted distributions were constructed by repeatedly shuffling the indices of both encoding and retrieval RDMs (100 iterations per subject), correlating the permuted RDM pairs, and averaging across permutations to yield a subject-level null estimate. Both observed and permuted values were compared using paired t-tests across subjects ($p < .001$; corrected for multiple comparisons using Benjamini–Hochberg method (Benjamini & Hochberg, 1995)).

### A.7.2 DECODING AND GEOMETRY ANALYSIS

**Decoding analysis.** For decoding analysis, we evaluated how well the obtained fMRI beta weights from different WM phases could classify the stimulus identity (8 subclasses (Gifford et al., 2025)). For each ROI and subject, we selected the top $k = 100$ voxels using univariate ANOVA ($f$-test) of stimuli class on the encoding data, and used the same subset in retrieval. In parallel, we extracted activations from models, separately for encoding and retrieval phases, respectively for the project and hidden layer.

For each ROI or model layer, we trained linear decoders to classify 8 image subclasses from extracted activity. To prevent the classifier from being dominated by high-frequency classes, we reweighted the loss inversely to class frequency during the training, and tested the decoding performance using the average of per-class recall accuracy. The chance level of the decoding accuracy is 0.125.

We used one-vs-rest linear SVMs with L2 regularization (C=1.0). The fitted decoders yield weight matrices $W_{\text{enc}}, W_{\text{ret}} \in \mathbb{R}^{k \times C}$, where $k = 100$ is the number of selected features and $C = 8$ is the number of stimulus classes. Each column of $W$ represents the normal vector defining the separating hyperplane for a class.

**Procrustes analysis and subspace alignment**. We normalized decoder weights columnwise and extracted orthonormal bases via SVD: $W'_{\text{enc}}$, $W'_{\text{ret}} \in \mathbb{R}^{k \times r}$, where $r = \min\left(\text{rank}(W_{\text{enc}}), \text{rank}(W_{\text{ret}})\right)$. In our case, $r = 8$.

We then estimated the orthogonal matrix $R \in \mathbb{R}^{r \times r}$ that best aligns encoding to retrieval subspaces by solving the orthogonal Procrustes problem (Gower, 1975):

$$R^{\star} = \arg \min_{R \in O(r)} \|W'_{\text{enc}} R - W'_{\text{ret}}\|_F.$$

The solution is given by $R^{\star} = UV^{\top}$, where $U\Sigma V^{\top}$ is the singular value decomposition of $W'_{\text{ret}}{}^{\top} W'_{\text{enc}}$. We used this singular values $\sigma_i$ of the $W'_{\text{ret}}{}^{\top} W'_{\text{enc}}$ to define the cosines of principal angles:

$$\cos \theta_i = \sigma_i, \quad \theta_i = \arccos(\sigma_i),$$

yielding an ordered sequence of angles with increasing degrees $0 \leq \theta_1 \leq \cdots \leq \theta_r \leq 90°$ that quantify alignment between subspaces.

For each ROI, we averaged principal angles across subjects to obtain an averaged canonical spectrum of rotation $\{\bar{\theta}_i\}$. For each model layer, we computed the corresponding spectrum of rotation from decoder weights. We then quantified brain–model similarity using the mean absolute deviation (MAD) between brain and model angles over the first $n$ axes:

$$\Delta_{\text{ROI}, \ell} = \frac{1}{n} \sum_{i=1}^{n} |\bar{\theta}_i^{(\text{ROI})} - \theta_i^{(\ell)}|,$$

where $\ell$ indexes the model layer. Setting $n$ gives the the average rotation degree of the top $n$ shared axes between encoding and retrieval subspaces. We report the differences in the rotation degree $\Delta_{\mathrm{ROI},\ell}$ between brain and model when $n = 1$ and $n = 3$, as in the top one and top three axes of space. Lower values indicate greater alignment between encoding–retrieval rotations in the brain ROI and the model layer $\ell$.

**Cross-decoding validation of rotation.** To test whether retrieval decision geometry can be explained as a rigid rotation of the encoding geometry, we performed a cross-decoding analysis in the learned subspaces. For each ROI and subject, we formed orthonormal bases for the encoding and retrieval decision subspaces, $U_e \in \mathbb{R}^{p \times d}$ and $U_r \in \mathbb{R}^{p \times d}$. Axes were *ranked by increasing principal angle* (largest cosine first), and the top-$k$ axes were retained to give $U_{e,k}$ and $U_{r,k}$ ($k = 1, \ldots, d$). The orthogonal Procrustes rotation aligning these $k$-dimensional subspaces was

$$R_k \;=\; \arg \min_{R \in O(k)} \|U_{e,k} R - U_{r,k}\|_F \,.$$

All preprocessing (feature selection, standardization) and the linear multi-class classifier were fit *only on encoding* to avoid information leakage. Let $X_e, X_r \in \mathbb{R}^{n \times p}$ be encoding and retrieval data. We trained a classifier $f$ on encoding-subspace coordinates

$$Z_e \;=\; X_e \, U_{e,k},$$

and then evaluated the *same* classifier on retrieval under two mappings:

$$\text{No rotation:} \quad Z_r^{\mathrm{naive}} \;=\; X_r \, U_{e,k} \qquad \text{Rotated:} \quad Z_r^{\mathrm{rot}} \;=\; X_r \, U_{r,k} \, R_k^\top \,.$$

The outcome for each $k$ was retrieval accuracy with and without rotation; we defined the gain as

$$\Delta_k \;=\; \mathrm{acc}_{\mathrm{rot},k} \;-\; \mathrm{acc}_{\mathrm{naive},k}.$$

For each ROI, $\Delta_k$ was computed per subject and then averaged across subjects to yield an ROI-level mean gain. To summarize "on average across ROIs," we took the mean of the ROI-level $\Delta_k$ values at each $k$. Statistical inference tested whether the across-ROI gain was above 0 using one-sided one-sample t-tests (a priori directional hypothesis that rotation improved cross-decoding).

## A.8 SUPPLEMENTARY RESULTS

### A.8.1 DECODERS TRAINED AND TESTED ON THE RETRIEVAL PHASE

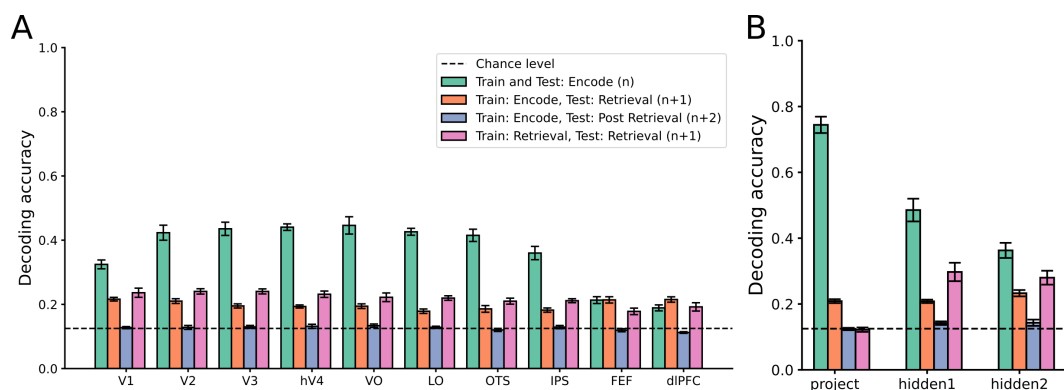

Figure A1: Decoding accuracies with decoders trained and tested on the retrieval phase. (A) Results of each human ROI. Error bars represent ± s.e.m. across 8 subjects. (B) Results of each model's layer. Error bars represent ± 1 s.e.m. across 12 models.

### A.8.2 RSA AND DECODING ANALYSIS RESULTS WITH CONTROL REGION PRIMARY AUDITORY CORTEX (A1)

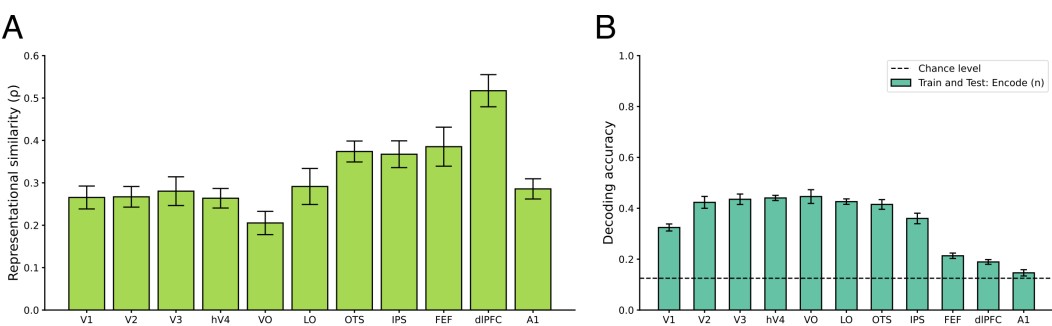

Figure A2: Results with control ROI A1. (A) Similarity scores (Spearman's $\rho$) of encoding and retrieval RDMs for each human ROI. Error bars represent ± 1 s.e.m across 8 subjects. (B) Decoding accuracies of the decoder trained and tested on the encoding phase. Error bars represent ± 1 s.e.m across 8 subjects.

### A.8.3   CENTERED KERNEL ALIGNMENT (CKA) RESULTS WITH RBF KERNEL

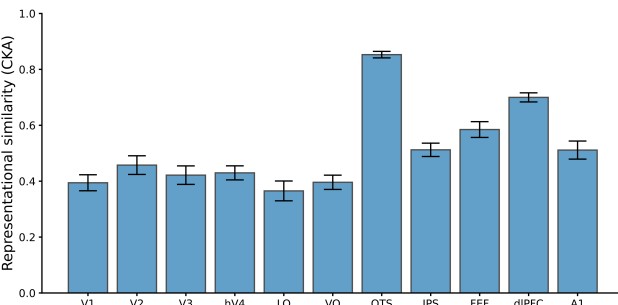

Figure A3: RBF kernel-based CKA results with control ROI A1. Error bars represent ± 1 s.e.m across subjects. The CKA RBF kernel parameter was set as 0.5. The parameters reflect the fraction of the median Euclidean distance used as $\sigma$.

### A.8.4   2-BACK TASK RESULTS

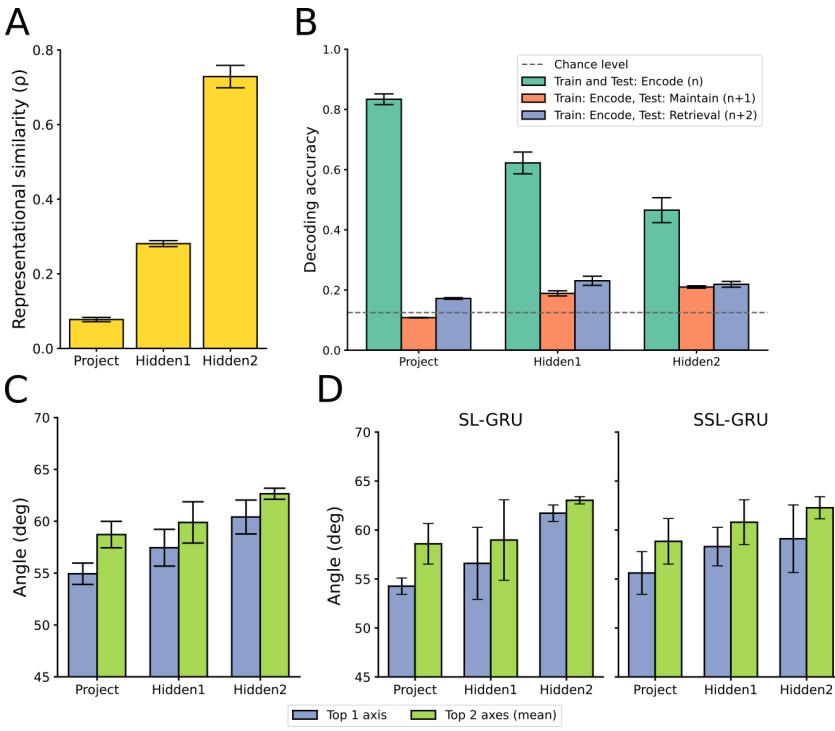

Figure A4: Analysis results with 2-back task models. Note that GRU was used as the recurrent module. (A) Similarity scores (Spearman's $\rho$) of encoding and retrieval RDMs for each model layer. Error bars represent ± 1 s.e.m across 4 models. (B) Decoding accuracies of each model's layer. Error bars represent ± 1 s.e.m across models. (C) Rotation angles of the top-1 axis and top-2 axes (mean) from encoding to retrieval in each layer of the models. Error bars represent ± 1 s.e.m across 4 models. (D) Group average rotation angles of the top-1 and top-2 axes (mean) from encoding to retrieval in each layer of the models. SL denotes supervised, SSL denotes self-supervised. Error bars represent ± 1 s.e.m across models with varying encoder architectures.

### A.8.5 MODELS WITH FINE-TUNED ENCODERS RESULTS

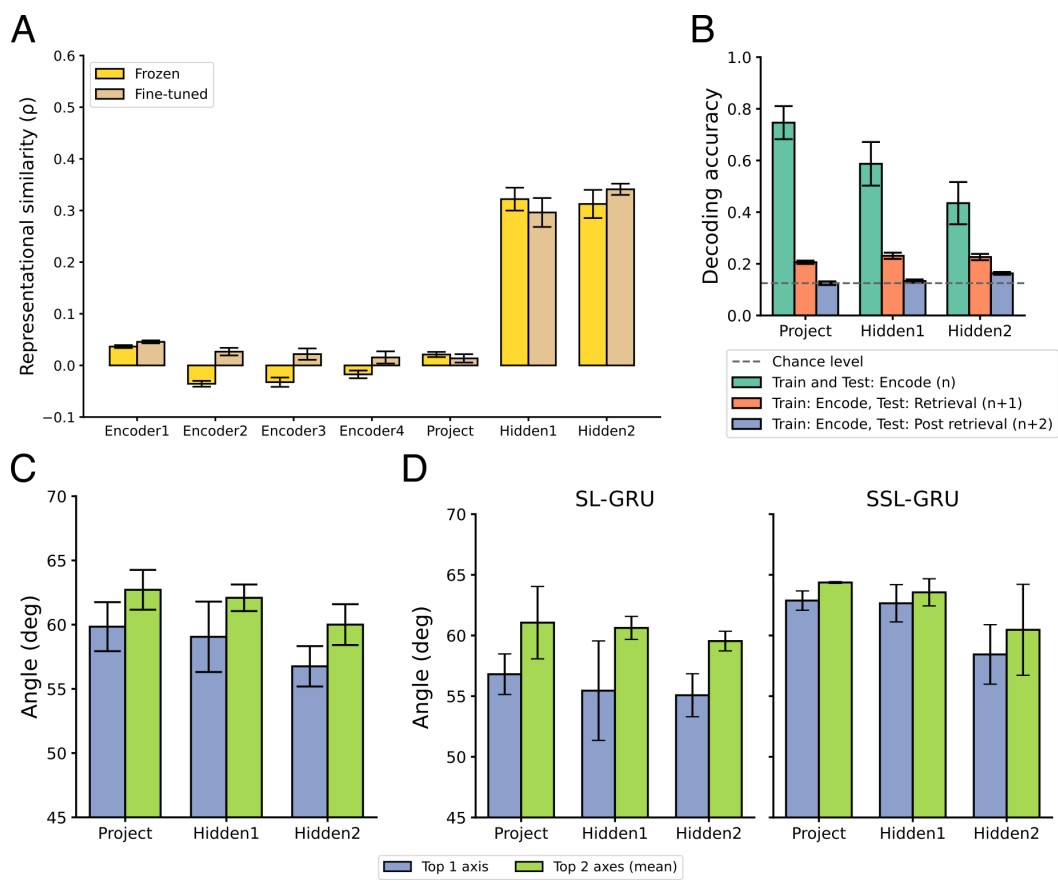

Figure A5: Analysis results of models with fine-tuned encoders. Note that GRU was used as the recurrent module for fine-tuned models. (A) Similarity scores (Spearman's $\rho$) of encoding and retrieval RDMs per model layer for both models with frozen encoders and with fine-tuned encoders. Error bars represent ± 1 s.e.m across models. (B) Decoding accuracies of each model's layer. Error bars represent ± 1 s.e.m across models. (C) Rotation angles of the top-1 axis and top-2 axes (mean) from encoding to retrieval in each layer of the models. Error bars represent ± 1 s.e.m across 4 models. (D) Group average rotation angles of the top-1 and top-2 axes (mean) from encoding to retrieval in each layer of the models. SL denotes supervised, SSL denotes self-supervised. Error bars represent ± 1 s.e.m across models with varying encoder architectures.

A.8.6 ADDITIONAL RSA RESULTS

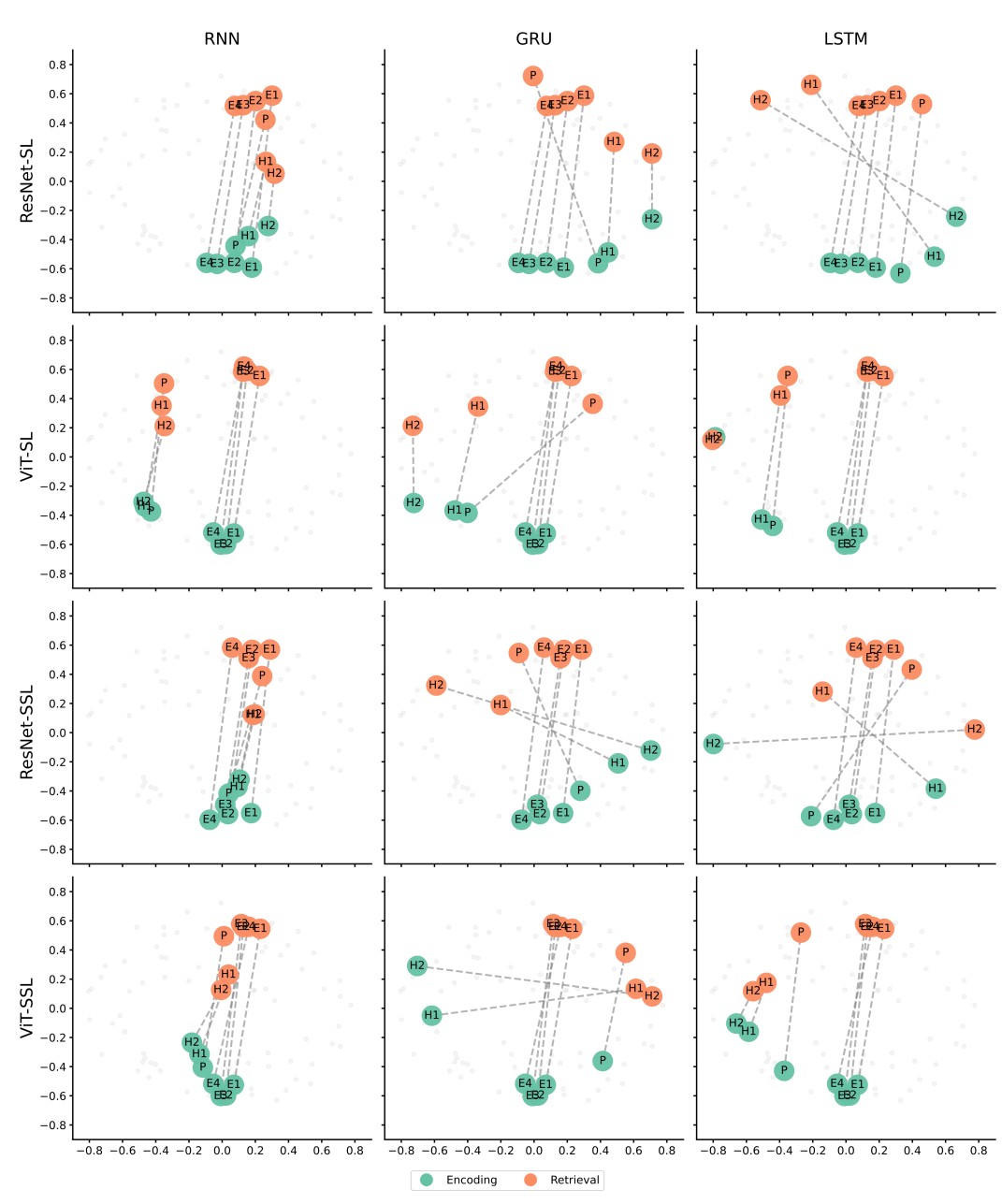

Figure A6: MDS visual-path of the dissimilarity matrix based on RDMs of encoding and retrieval representations for each model layer. E denotes frozen weight encoder layers, P denotes the project layer, and H denotes hidden layers. The number following the letter denotes the order of that layer. Note that across all the models, the similarity scores across encoding to retrieval phases were higher in 2 hidden layers compared to the project layer (FDR post-hoc paired t-tests, $p < .001$).

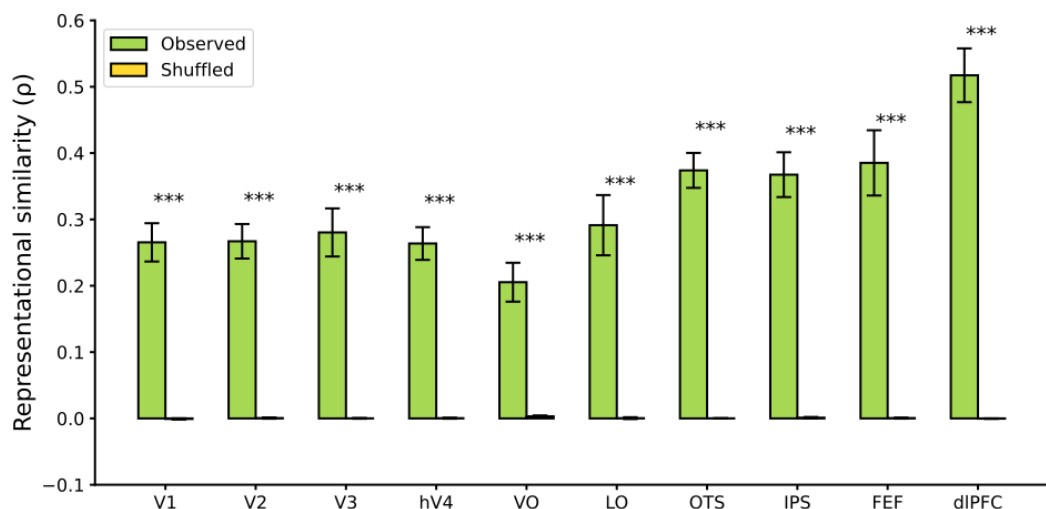

Figure A7: Shuffled RDM representational similarity results. Repeated-measures ANOVA showed significant effects of condition and ROI, and their interaction ($p < .001$). FDR post-hoc paired t-tests confirmed robust differences between observed and shuffled conditions across all ROIs. $p < .05$: *; $p < .01$: **; $p < .001$: ***.

