# OpenReview forum: "Hierarchical Representational Transformations of Working Memory in Brains and Machines"
_ICLR.cc/2026/Conference — Submitted to ICLR 2026_

### Official Review · Reviewer_ts76 · 2025-10-15

**Soundness:** 3
**Presentation:** 3
**Contribution:** 3
**Rating:** 4
**Confidence:** 3

**Summary:**

This work explores how working memory representations transform between encoding and retrieval phases in both human brains and artificial recurrent networks. Using 7T fMRI data from the Natural Scenes Dataset and RNNs trained on a 1-back task, the authors compare representational similarity, cross-decoding, and subspace rotation metrics to identify whether transformations are stable, rotational, or dynamic. They find that early visual regions show strong transformations while higher cortical regions are more stable, and that gated RNNs with supervised encoders best recapitulate these patterns.

**Strengths:**

The integration of high-field fMRI with modern deep networks to probe representational dynamics across biological and artificial systems is timely and relevant. The framing around WM encoding–retrieval transformations is novel and theoretically motivated.

I liked the combination of RSA, cross-decoding, and geometric rotation analysis. All together these provide a multifaceted characterization of representational transformation.

The paper is clearly structured, includes well-labeled figures, and discusses limitations and reproducibility carefully.

It's an interesting use of the NSD dataset. However I was slighlty disappointed by the subselection of trials (one the main advantage of the NSD is his size, I understand keeping the experiments cheap and computationally feasible but I felt some potential is wasted here just by not using more data that are available)

**Weaknesses:**

While the analyses are well executed, the conceptual novelty is somewhat limited. The notion of mixed stable and dynamic subspaces in WM is well established (e.g., Stokes et al. 2013; Murray et al. 2017 and others). The contribution lies mainly in replicating this finding in 7T fMRI and showing partial correspondence with RNNs.

Only one participant from NSD is shown for visualization, and the main analyses appear averaged across subjects. Inter-subject variability, cross-validation robustness, or the reproducibility of representational gradients across individuals are not reported. Given the small number of participants (n=8), the statistical reliability of subspace angles and cross-decoding should be demonstrated per subject or with permutation testing.

While RSA and rotation-angle analysis are intuitive, they reduce high-dimensional geometry to single scalar measures. Methods like Procrustes alignment, representational connectivity analysis, or canonical correlation analysis (CCA) could better capture transformations. The “top-1 angle” approach seems ad hoc and may not robustly separate rotational vs. non-rotational effects.

The encoders are frozen, which limits the capacity for task-specific representational adaptation that occurs in the human visual system during WM tasks. It was proven several times in audio (Brain-tuning) and during the Algonauts challenge 2025 (See CCN 2025) that fine-tuning these systems could improve the quality of learned representations, potentially strenghtening the results.

RNNs are tipically used in neuroscience, but authors could maybe want to try an attention based model? It's a different type of bias and therefore a different assumption.

**Questions:**

You report that supervised encoders and gated RNNs show “brain-like” hierarchical gradients of representational stability, but this comparison is largely qualitative. Could you quantify the correspondence between cortical ROIs and model layers—for example using cross-system RSA, centered-kernel alignment (CKA), or Procrustes distance—so that the claimed alignment can be statistically evaluated rather than inferred descriptively?

The rotation-angle analysis focuses on the top-1 or top-2 principal axes, that usually captures the first moments of the distribution. In other tasks such as image retrieval from brain activity or decoding it seems to be quite relevant to account for more PCs. How sensitive are these results to dimensionality choices and noise structure? Could you comment on that?

Supervised encoders appear to match cortical gradients better than self-supervised ones, yet biological learning is unlikely to be explicitly supervised. On the other hand, for encoding, decoding or RSA between brain and models it was proven multiple times that unsupervised model reach better scores (See "What can 5.17 billion regression fits tell us about artificial models of the human visual system?" for example, or Algonauts 2023/2025 results or similar literature, this result is pretty consistent)
Could you dissect what specific representational property of supervised training (e.g., category clustering, linear separability, or contrastive invariance loss) drives this alignment? Is the difference preserved after controlling for feature discriminability or category structure?

How do you expect your results to change if you use more data? This was a quite unexpected choice in your work.

---

> ### Author Response · Authors · 2025-12-04
> **Rebuttal responses to Reviewer ts76 (1/2)**
>
> We thank the reviewers for their constructive comments. We appreciate the reviewer’s positive comments highlighting the clear motivation, the novelty and clarity of the work, and especially for recognizing our analysis approach that provides multi-angle examinations on the research topic.
>
> To address issues raised, in the revision, we included additional discussions in conceptual novelty, data and analyses choice, and the impact of training objectives. We also added additional statistical tests and high-dimensional analysis, and conducted experiments with fine-tuned encoder models. Below we addressed each comment by the reviewer.
>
> **1. Conceptual novelty**
>
> To clarify, previous studies claiming a coexistence of stable and dynamic coding are based on single-ROI (prefrontal) recordings in monkeys, with a special stress on how the neural tuning profiles in prefrontal cortex adapt to accommodate changes in behavioral tasks. Here we consider various human ROIs involved in visual WM processing, which parallel the variation of layers in neural networks. The central conclusion – the dynamic to stable coding across region/layer hierarchy in humans and models – is hence novel to the previous literature.
>
> Beyond the novelty listed in the manuscript such as using naturalistic stimuli and linking humans and models in the Related work section, we will add this note to it.
>
> **2. ROI visualization and Permutation test**
>
> It is common to use an example subject for brain ROI visualization as in **Figure 1D**,  but all the human results we reported in **Figure 2–4** used the ROI activations across all participants (mean ± 1 s.e.m).
>
> We considered inter-subject variability by formally reporting the statistical tests to the calculated transformation metrics —RSA, decoding accuracy, and rotation angles— across the participants. We agree that due to the small number of participants (n=8), non-parametric or permutation tests could be a better option. We updated the statistical tests with the permutation version in our revised manuscripts. All the updated results reproduced the same results supporting our conclusion that: humans and models showed a mixed strategy of representation transformation between WM encoding and retrieval.
>
> **3. High-dimensional geometry analysis**
>
> We conducted the RDF CKA analysis, which operates in a high-dimensional (possibly infinite-dimensional) feature space defined by the kernel. With more details in response point 2 to Reviewer ppZL, RDF CKA results also indicated a more dynamic WM coding mechanism in the low- and mid-level visual areas, and a relatively stable coding in the frontal regions.
>
> **4. The choice of top-1 and top-2 angle**
>
> We discussed the rationales behind the choice of the top-1 and top-2 axes angles in response point 6 to Reviewer mji9. We agree that the rotation results may be sensitive to the selection of the dimension (PC): only the top 1–2 dimensions exhibit meaningful rotation between encoding to retrieval representation, whereas the remaining dimensions reflect non-rotational changes (e.g., shifts in relative representational distances, scaling, or shearing).
>
> **5. Fine-tuned encoder models**
>
> We added the results from models with fine-tuned encoders, see response point 3 to Reviewer ppZL. In short, the choice of a fine-tune or frozen encoder doesn’t impact our central finding on a dynamic WM coding in the lower project layer and a more stable representation in the higher layers across WM encoding to retrieval phases.
>
> **(Responses continued in the following comment)**

---

> > ### Author Response · Authors · 2025-12-04
> > **Rebuttal responses to Reviewer ts76 (2/2)**
> >
> > **6. Attention-based models**
> >
> > We appreciate the suggestion. In this work, we systematically evaluated 12 model configurations spanning 2 encoder architectures (CNN, ViT), 2 encoder learning objectives (supervised, self-supervised), and 3 recurrent modules (vanilla RNN, GRU, LSTM). These choices were motivated by their widespread use in neuroscience-inspired modeling. However, we don't think that our results strongly hinge on the specific model architecture choice: The main conclusion (larger WM transformation in earlier layers and more stable code in later layers) were supported with results averaged across 12 models (**Figure 2–3** and **Figure 4B**).
> >
> > Modern attention-based models are also commonly used to model memory processes, while more focus is on episodic or long-term memory (Dong et al 2025; Li et al 2024; Pink et al 2024) due to its strong ability to retrieve information from long context (e.g., 128k tokens for GPT-4-128L), which is outside the domain of WM. The RNN, however, is widely used as a model of working memory, due to its ability to maintain information over time and generate persistent dynamics in parietal / frontal regions (e.g., Compte et al., 2000; Wang, 1999; Bouchacourt & Buschman, 2019; Wimmer et al., 2014; Esnaola-Acebes et al., 2022; Yang & Wang, 2020; Wang, 2021). Our task is a simple, short temporal WM task, so we only considered RNNs in the current work.
> >
> > Recent studies proposed key-value based memory theory (Gershman et al 2025) which distinguish representations used for memory (values) and for retrieval (keys). We are interested in assessing whether humans and models adopt this strategy in WM processing, and consider attention-based models as a promising future direction.
> >
> > **7. Cross-system brain-model alignment**
> >
> > To clarify, our study does not evaluate models based on the predictivity of brain responses using model representation during the task, which is the central question of previous studies of brain-model alignment (e.g. Conwell et al 2024; Wang et al 2023). Rather, we ask whether models—regardless of training objective—exhibit similar representational transformations across WM encoding to the retrieval phase. This leads to a different alignment criterion in our work.
> >
> > Crucially, our main theoretical finding—that earlier layers show larger representational transformations while later layers show more stable codes, mirroring human neural strategy in WM—is robust across all 12 models regardless of architectures or training objectives (**Figures 2–3**). RSA results capture rotation-invariant geometry, while decoding analyses capture both rotational and non-rotational transformations, providing convergent evidence.
> >
> > To directly address this question, we conducted RSA analysis comparing the model representation and brain activation in the same WM phase. Overall, the RSA scores remained low (~0.1) across model projects to hidden layers, brain regions, and WM phases. Previous studies showed merely adding a recurrent module to encoders showed worse brain activation predictivity than only the encoder models (Al-Karkari et al, 2025). Additionally, the training goal of the models was to perform the 1-back task, so the emergence of brain predictivity or the human-like transformation we reported were not expected. Therefore, human brains and models holding distinctive representational structures could still show similar mechanisms of transformation in the working memory processing.
> >
> > **8. Self-supervised and supervised learning**
> >
> > We discussed how training objectives (SL or SSL) affects model’s alignment with human data in the response point 2 to reviewer 1p9M.
> >
> > **9. Choice of the data**
> >
> > To clarify, we tried to exploit the data available: the models were trained with the NSD-core but tested with the full NSD-synthetic data. Though we agree that a larger or more naturalistic 1-back task dataset would strongly improve our paper. Additionally, it's crucial for this study to directly assess WM transformation in humans and models with the same dataset, so we could only test the models with NSD-synthetic images not with NSD-core.
> >
> > For the training phase with NSD-core, we made sure all 80 categories defined in COCO are included and we balanced the number of images in each class in each training set. The model training consisted of 40 runs while the testing included only 4 runs.
> >
> > To generalize our findings, we also trained models with NSD-core and tested with multiple randomly generated sets with NSD-synthetic, the results of which were reported in response point 3 to Reviewer mji9.
> >
> > We also discussed the generalization from a 1-back task to a broader scope in response point 8 to Reviewer mji9, and we added model results in 2-back task in response point 1 to Reviewer ppZL.

---

### Official Review · Reviewer_ppZL · 2025-10-30

**Soundness:** 2
**Presentation:** 2
**Contribution:** 2
**Rating:** 4
**Confidence:** 4

**Summary:**

This paper investigates how working memory (WM) representations transform between encoding and retrieval in both the human brain and recurrent neural networks. Using 7T fMRI data from the NSD-synthetic dataset and RNN models trained on a 1-back image task, the authors analyze representational similarity, cross-decoding, and geometric subspace rotations across cortical regions and model layers. They report that early visual regions undergo stronger transformations while higher-order prefrontal regions remain stable, and that gated architectures (LSTM, GRU) and supervised encoders best align with this hierarchical pattern.

**Strengths:**

This study bridges neuroscience and machine learning in a clear and structured way. The use of high-resolution fMRI with model-based analyses provides a rigorous framework for comparing representational dynamics. The combination of RSA, cross-decoding, and subspace-geometry analysis gives convergent evidence for hierarchical differences between dynamic and stable WM codes. The inclusion of both supervised and self-supervised visual encoders also adds an informative comparison for understanding how training objectives shape representational transformations.

**Weaknesses:**

The main limitation is that both the behavioral and modeling aspects are confined to a single, highly specific task—the visual 1-back paradigm. This narrow scope restricts the generality of the conclusions about “working memory mechanisms” in either brains or machines. The framework does not address whether the same transformation principles would hold across different WM domains (e.g., auditory, spatial, or rule-based memory) or task complexities.

Methodologically, the model design is constrained by strong assumptions: the frozen encoder architecture and fixed training regime prevent adaptive representational learning, making the observed “alignment” largely descriptive rather than mechanistic. The subspace rotation metric, while intuitive, simplifies nonlinear representational changes and risks over-interpreting correlations as shared mechanisms. Moreover, analyses such as RSA and cross-decoding rely on correlation-based similarity, which cannot disambiguate dimensional scaling, feature reweighting, or other geometric deformations.

Overall, while the results are well presented, the modeling approach feels confirmatory—testing pre-defined correspondences on a single dataset—rather than probing how flexible task dynamics or alternative objectives might yield convergent or divergent brain-like transformations.

**Questions:**

Could the authors test their claims on a broader range of WM tasks or stimuli to evaluate whether the observed hierarchical pattern generalizes beyond 1-back visual memory?

How much of the model–brain “alignment” arises from architectural choices (e.g., frozen encoders, layer normalization) versus learning dynamics?

Would fine-tuning the encoder jointly with the recurrent module change the geometry of encoding–retrieval subspaces?

The rotation-based metric assumes linear subspace alignment — have the authors considered nonlinear manifold alignment or mutual-information–based measures to test representational overlap?

Since the recurrent models are trained on synthetic data with limited variability, could the results be partly driven by dataset idiosyncrasies rather than genuine hierarchical transformation?

---

> ### Author Response · Authors · 2025-12-04
> **Rebuttal responses to Reviewer ppZL (1/2)**
>
> We thank the reviewer for their constructive comments, and for recognizing the importance, rigorous analyses, and convergent evidence of the work. During rebuttal, we added clarifications on the training dataset and our non-confirmatory approach. We also added additional analyses showing our observations can be generalized to non-linear methods, the 2-back task, and fine-tuned encoder models. Below we address each comment by the reviewer.
>
> **1. Observed results can be generalized to a 2-back task**
>
> We appreciate the reviewer’s comment regarding the generality of our findings. The n-back task is widely used to investigate the memory mechanisms in both humans (EEG: Wan et al 2022; Wan et al 2022), and non-human primates (Meyer and Rust 2018; Mehrpour et al 2021; Jannuzi et al 2025; DiRisio et al 2025). For example, the 1-back / n-back task was used to investigate mechanisms of WM transformation in models (Lei et al 2024), priority based representational changes (Wan et al 2020; Wan et al 2022), etc. More literature on the generalization of 1-back task see response point 8 to reviewer mji9.
>
> To directly address the reviewer’s concern regarding task complexity, we extended our modeling analysis to a 2-back task (shown in manuscript **Supplementary Figure A4**). With a similar procedure, we trained the models to judge whether the current image matched the stimulus presented two steps prior, thereby increasing the memory load and temporal horizon. We then applied the exact same analysis pipeline (RSA, decoding, and geometry) to investigate the representational transformation.
>
> The RSA scores monotonically increased across model layers. This reveals that the change from dynamic coding (in the projection layer) to stable coding (in the deeper hidden layers) was preserved in the 2-back task.
>
> As for the decoding results, the decoder trained in encoding phase (n) and tested in the retrieval phase (now n+2, instead of n+1) for all layers showed above the chance accuracy, suggesting a reliable generalization of the encoded information to the retrieval phase. However, the decoders trained in encoding (n) and tested in the maintenance (n+1) achieved above the chance accuracy only in hidden layers, suggesting the unique role of online memory maintenance of the recurrent module.
>
> Lastly, the rotation angle in the 2-back task increased as  model layers progress, opposed to the decreasing angles across layers in the 1-back task, indicating an increased rotational transformation in the later layer. The different results may be due to the increased online storage demand in hidden layers during the 2-back tasks, which requires a clear separation of the concurrently encoded, maintained, and retrieved information in each subspace.
>
> In summary, we observed consistent evidence that lower layers experienced more non-rotational transformation compared to the later layer, regardless of the task difficulty.
>
> **2. Non-linear method**
>
> To capture the non-linear manifold structure similarity between the representation of WM encoding and retrieval phase that can’t be captured by RSA or linear decoding, we added the non-linear kernel (RBF) CKA analysis. As shown in **Supplementary Figure A3**, the RBF CKA similarity score was higher in the frontal region dlPFC, while early to mid level visual areas exhibited lower similarity score, similar to RSA results. However, OTS exhibited higher similarity in CKA than what we observed in RSA. One possible explanation is that encoding and retrieval in OTS engage a largely shared representational subspace, which could be quantified by CKA, but not the arrangement of individual stimuli within this space (i.e., which stimuli are relatively closer or further from each other) – captured by RSA. In other words, OTS may reshape the geometry of stimulus relationships by linearly re-weighting a shared feature space.
>
> We also added the models with the same architecture but fine-tune encoders (see response point 3 to reviewer ppZL), and also 2-back task results (see response point 1 to reviewer ppZL). As a summary, those results exhibited convergent evidence supporting the central contribution of the current work—a dynamic earlier region showed a larger WM transformation across phases while the later regions use a more stable WM code.
>
> **(Responses continued in the following comment)**

---

> ### Author Response · Authors · 2025-12-04
> **Rebuttal responses to Reviewer ppZL (2/2)**
>
> **3. Effects can be generalized to fine-tuned encoder models**
>
> Thank you for the question. A sensory->cognitive structure of models was widely used to explain WM mechanisms (Bouchacourt & Buschman 2019; Yang et al 2024; Xie et al 2023; Lei et al 2024). In our architecture, the encoder (ResNet/ViT) is designed to provide a sensory perception while the recurrent module is intended for higher-order memory processing. This encoder-recurrent combination has been adopted to resemble sensory->cognitive structure in human WM (Xie et al 2023; Lei et al 2024), and it’s a common practice for the previous studies to freeze the encoder part that is pretrained with large natural image dataset.
>
> To directly test this, we trained additional models with identical architectures but with their encoders fine-tuned. After initializing the encoder with ImageNet-pretrained weights, we allowed the encoder layers—previously reported as frozen—to update during training (results in **Supplementary Figure A5**).  As for RSA scores, fully fine-tune models showed similar relative representational distances with the frozen encoder models across project to the hidden layers. Additionally, fine-tuned encoder layers showed low similarity scores between WM encoding to retrieval phases in the encoder layers, matching the hypothesis that encoder layers were not involved in memory processes. Based on the decoding and the rotation analysis results, fine-tuned models exhibited highly similar WM transformations strategies as their frozen encoder counterparts reported in the manuscript.
>
> Overall, the choice of a fine-tune or frozen encoder doesn’t impact our central finding on a dynamic WM coding in the lower project layer and a more stable representation in the higher layers across WM encoding to retrieval phases.
>
> **4. Non-confirmatory approach**
>
> To clarify, we don’t have a priori assumptions of what WM transformation we would observe, and how it might change across brain regions. We are constrained by the availability of high-resolution human fMRI neural dataset that uses a n-back task, and NSD-synthic is what we found. We'd be happy to generalize to more tasks and datasets if it’s available. We also don't have a priori hypotheses on what kind of recurrent units, or training objective would yield a more human-like WM coding strategy.
>
> Additionally we added the models with the same architecture but fine-tuned encoders (see response point 3 to reviewer ppZL), 2-back task results (see response point 1 to reviewer ppZL), and non-linear CKA results (see response point 2 to reviewer ppZL). As a summary, those results exhibited convergent evidence supporting our central conclusion—a dynamic earlier region showed a larger WM transformation across phases while the later regions use a more stable WM code.
>
> **5. Training set clarification**
>
> To clarify, the models were trained with the NSD-core natural scene images (Allen et al 2022; more details in Appendix 3.5). After training, the models were tested with the out-of-distribution synthetic images in the NSD-synthetic (Gifford et al 2025). We will add detailed descriptions in the revised manuscript section 4.1.
>
> We also trained models to perform a 2-back task (see response point 1 to reviewer ppZL). Compared to the 1-back task, models in the 2-back task also exhibited stronger hierarchical changes regarding the WM transformation across phases.

---

### Official Review · Reviewer_mji9 · 2025-11-02

**Soundness:** 1
**Presentation:** 3
**Contribution:** 2
**Rating:** 2
**Confidence:** 3

**Summary:**

This study investigates differences in representations across the encoding and retrieval phases during a 1-back working memory task. Using the natural scenes dataset, they perform a series of parallel analyses on fMRI activity from working memory associated regions of the human brain and activations from two visual encoders (ResNet & ViT) and layers of different recurrent neural network architectures (RNN, GRU, LSTM). They find that the encoding and retrieval phases of working memory have partially overlapping representations, but that they follow rotational and non-rotational transformations across the visual, mid-level, and frontal regions. These findings were somewhat replicated in the recurrent neural network models, with supervised training regimes and gating architectures capturing the most brain-like representations.

**Strengths:**

- The analysis into WM representation dynamics of the 1-back task was of high depth (combined RSA, cross-decoding, and geometry)
- They find an interesting contradiction to previous findings (https://pmc.ncbi.nlm.nih.gov/articles/PMC7826371/) on supervision and the learning of brain-like representations.
- The methods and results of the paper were presented in a clear manner
- They perform strong statistical tests to validate results

**Weaknesses:**

Related to Soundness:
- I have a hard time imagining how the brain can solve any n-back task with a stable memory representation. Given that in any n-back task new stimuli are constantly coming in, I am struggling to see how any algorithm can solve this task without a dynamic neural code. In the absence of such candidate, I had to retreat to assume the possibility of other scenarios being at work that may explain the high RDM similarity in dlPFC. At the top of this list is that this area is either not involved in solving this task altogether or that only a portion of this region is involved. If some of these regions are not involved in solving this task, I would expect that the activity in those voxels to be random and as a result the SNR to be lower. A helpful way to give some insights into this would be to check the SNR within different ROIs. Related to this point, fig3 results shows that the E->E decoding accuracy is much lower in FEF and dlPFC compared to other areas. This may indicate that the measurement noise is generally higher in those areas. However, it could also indicate that the information is not encoded in these areas in a linearly decodable manner.

- It is unclear whether multiple instances of each model were tested or not. I didn't notice any mention of having multiple seeds for each model. Error bars are non-existent in Fig1 plots and in other figures where they exist, it is unclear how they were computed.
- An alternative view of the relatively high similarity in associational areas is that they may show relatively less variation to stimuli in general.  For example, the activity in some or many voxels may remain unchanged in response to changing stimuli because they are not being involved in processing 1-back task. This may be interpreted as evidence of a stable code while in reality parts of the network (i.e. voxels) may not be involved in performing the task altogether.
- Lei et al. 2024 had shown that the distribution of tasks on which a model is trained could have drastic effects on its learned representational geometry. This is an important factor that was not considered in the study and could potentially affect the interpretations drawn from the experiments.
- The model consists of 2 RNN layers. It's unclear how this architecture is perceived to map onto the brain. Why were two layers selected? How does that choice in general affect biological similarity? Both anatomically and representation-wise?
- why were only top-1 and top-2 axes used in fig4 analyses? I noticed a description of the Procrustes analysis in the extended methods. Is that used to perform the rotation analyses? Why wasn't it tried on all decoding directions and instead the angles of top1/2 principal angles are used?

Related to Presentation:
The paper is presented generally well; however, there are several areas that need improvements:
- Lines 177-178: Remove redundant “in addition to the project layer.”
- Lines 345-349: The statement “..the rotation angle between encoding and retrieval was smaller in the low-level regions (V2, V3) compared to the high-level frontal areas…” seems to contradict the findings for top-1 angles just mentioned above it “...reduction in rotations when moving from low-level visual areas to higher-level brain regions..”
- which model type is shown in fig2?
- what were the decoders in fig3 trained to do? It’s unclear what the decoding task is
- line 321: while the difference between E-E and E-R accuracies is lower in these two areas, the E-E accuracy is substantially lower than other areas. It is misleading to interpret that as evidence of “better generalization”
- line 324-325: I’m not sure which result is used to back this statement. Fig 3b shows a significant drop in performance between the E-E and E-R cases.

Related to Contribution:
- The paper's contributions and scientific impact are diminished by the limited working memory task space (1-back object matching only). While the methods and results for this specific task are interesting, general conclusions about how working memory encoding and retrieval representations unfold across time cannot be drawn from them. Ideally, this kind of analysis would have covered a wider range of tasks, as the similar Lei et al. 2025 paper did (9 tasks), as mentioned in the introduction. While NSD is a limiting factor, and that no other broad visual working memory fMRI datasets exist yet; nonetheless, the contributions are still diminished as a result.

- Including more modern architectures such as state-space models would have improved the breadth of the findings
- A more direct way of comparing similarity between the model representations and the brain would have been to measure predictability of different brain ROIs from each model type using linear regression or similar methods.

**Questions:**

Why was the use of only 1-task not mentioned in the limitations? Do the authors feel this task paradigm is sufficient to support the broad claims on the nature of encoding-retrieval representations during WM? If so, why?

---

> ### Author Response · Authors · 2025-12-04
> **Rebuttal responses to Reviewer mji9 (1/3)**
>
> We thank the reviewer for the constructive comments. We appreciate the reviewer’s positive comments highlighting the depth of our methodology and rigorous statistical analyses. To address the issues raised, in the rebuttal, we added analyses of control region supporting the engagement of frontal regions, added multiple seed results, and clarifications of methodology choice, and generalized the observations to the 2-back task. Below we addressed each comment by the reviewer.
>
> **1. Frontal regions WM information can be reliably decoded**
>
> We thank the reviewer for the insightful comments. Indeed, the SNR in the frontal area can be lower than the early visual regions V1-hV4, which is in line with the human fMRI findings that reported lower decoding accuracy in frontal regions compared to the early visual areas during WM (Serences 2016; Curtis and D’Esposito 2003; Li and Curtis 2023; Christophel et al 2017; Emrich et al 2013). However, these studies also consistently reported the persistent activity in the frontal region during WM maintenance, and robust above-chance decoding, supporting that frontal regions are engaged in the WM process.
>
> We used an atlas from Glasser et al. (2016) to extract one control ROIs that are regarded as not involved in visual WM tasks: primary auditory cortex (A1). As shown in the manuscript **Supplementary Figure A2B**, with a permutation test, A1 showed decoding accuracy no different than chance (permutation $p > .05$, FDR-corrected). However, all the selected ROIs in the manuscript (V1-dlPFC) showed a higher decoding accuracy than the chance level ($p < .01$). Furthermore, FEF and dlPFC exhibited higher decoding accuracy than the control area A1 ($p < .01$).
>
> Thus, despite a lower decoding accuracy in frontal areas compared to other visual regions, FEF and dlPFC are actively engaged during WM processing and their activations can be meaningfully decoded.
>
> **2. The variation does not determine the similarity score**
>
> We agree that a low variation in the signal could possibly contribute to high representational similarity. To address this, we applied RSA to the control region A1 (results shown in **Supplementary Figure A2A**). We found that the RSA score of A1 is not different from V1-hV4, VO-LO (permutation $p > .05$, FDR corrected), and lower than frontal FEF and dlPFC ($p < .05$). Note that A1 doesn’t contain decodable information as we’ve shown in the previous response (Supplementary Figure A2B). Therefore, **a low decodability does not necessarily co-vary with a high RSA score.**
>
> Moreover, the RSA scores between WM encoding and retrieval phase representations in FEF (0.39) and dlPFC (0.52) don’t suggest a fully stable representation geometry across WM phases. While the RSA of dlPFC is higher than other regions (permutation p < .05, FDR corrected), the other frontal region FEF has no different RSA score compared to early to mid level regions (p > .05) except V2.
>
> Combined with the reliably above chance decoding of FEF and dlPFC during WM encoding and retrieval (response point 1 to Reviewer 1p9M and shown in Supplementary Figure A1A), both decoding and RSA results could support that frontal regions are involved during the WM process. Compared with early visual regions, dlPFC and FEF experienced less transformation across WM phases.
>
> **3. Multiple seed results and error bar descriptions**
>
> We updated **Figure 1C** results with models trained and tested with multiple seeds. Within each seed condition out of 5 seeds, we trained models with the same procedure as described in the manuscript (across 5 levels of match trial probability), and tested the models with multiple test sets generated with NSD-synthetic images. The bar shows the mean ± 1 s.e.m across averaged testing performance over all the testing sets of models across 4 encoder types.
>
> Thanks for the helpful suggestion on the errorbar, and now we added the clear description of errorbar calculation in each figure caption. For results Figure 2D, 3B, and 4B, the error bar is the standard error of all 12 models’ results. For results Figure 4C, the error bar is the standard error of each layer’s rotation angle across models with varying encoder architectures (ResNet or ViT).
>
> **4. The task distribution of the training set**
>
> The models in our work were trained with the naturalistic scene images in the NSD-core (Allen et al 2022), and tested with the synthetic images in the NSD-synthetic (Gifford et al 2025). Those synthetic images are out-of-distribution from the NSD-core based on fMRI responses (Figure 3a in Gifford et al 2025)
>
> Additionally, we agree that we are constrained by the availability of the high-quality fMRI WM dataset, and we will add this note to the limitation. We also discussed the generalizability of our conclusions to other WM operations in response point 8 to reviewer mji9, and reported the 2-back task model results in response point 1 to reviewer ppZL.
>
> **(Responses continued in the following comment)**

---

> ### Author Response · Authors · 2025-12-04
> **Rebuttal responses to Reviewer mji9 (2/3)**
>
> **5. How does the architecture choice map onto the brain**
>
> A sensory-cognitive structure of models was widely used to explain WM mechanisms (Bouchacourt & Buschman 2019; Yang et al 2024; Xie et al 2023; Lei et al 2024). The encoder (ResNet/ViT) is designed to provide a biologically motivated visual hierarchy corresponding to ventral areas for sensory perception (Yamins and DiCarlo 2016; Schrimpf et al 2018). The recurrent module is intended to model higher-order WM circuitry due to their ability to maintain information over time and generate persistent dynamics (e.g., Compte et al., 2000; Esnaola-Acebes et al., 2022; Yang & Wang, 2020; Wang, 2021). **A encoder-recurrent combination has been widely adopted to resemble sensory-cognitive structure in human WM** (Xie et al 2023; Lei et al 2024).
>
> Within the WM stage, we use 2 recurrent layers since many theories and models of WM (e.g., Wang 1999; Wimmer 2014; Bouchacourt & Buschman 2019; Kular et al 2025) posit that higher-order regions involve **multiple interacting subcircuits** supporting both flexible updating and stable maintenance. A single recurrent layer might be hard to separate these computational roles, while deeper architectures introduce unnecessary complexity.
>
> Empirically, the two RNN layers naturally differentiate in a way that mirrors human decoding results when trained and tested in the encoding phase: 1st hidden layer showed better decoding results compared to the 2nd layer—matching the human fMRI findings that during WM, early visual areas showed higher decoding accuracy than higher regions (Serences 2016; Curtis and D’Esposito 2003; Li and Curtis 2023; Christophel et al 2017).
>
> Thus, 2 recurrent layers are the **minimal and sufficient** architecture that (1) captures the diversity of transformations observed in higher-order WM regions, and (2) allows us to test how these WM computations align with brain data, while the encoder accounts for the lower-level visual hierarchy.
>
> **6. The choice of top-1 and -2 axes rotation angles**
>
> Yes, the rotation angles in Figure 4 were computed using the Procrustes-based procedure described in the appendix. Importantly, our goal was to quantify whether any part of the encoding to retrieval transformation can be meaningfully approximated as a rigid rotation. Procrustes analysis returns a full rotation matrix, but the higher-order angles rapidly saturate at ~90° across human ROIs. A 90° principal angle indicates either an orthogonal rotation or a complete non-rotational transformation.
>
> To determine which dimensions actually support transferable geometry, we performed the validation analysis described in Appendix A.4.2: training a decoder in the encoding phase and testing it on the retrieval phase with and without applying the derived rotation. Across ROIs, applying the rotation only improved generalization when restricted to the top 1–2 most aligned axes ($p < .05$), but not when including additional axes. This shows that **only the top 1–2 dimensions exhibit meaningful rotational encoding to retrieval transformation**, whereas the remaining dimensions reflect non-rotational changes (e.g., shifts in relative representational distances, scaling, or shearing).
>
> **7. Redundant and confusing parts in writing**
>
> Thank you for pointing out the redundant parts in writing, we will revise it in the latest version.
> - Yes you are correct, it should be “the rotation angle between encoding and retrieval was *larger* in the low-level regions (V2, V3) compared to the high-level frontal areas”.
>
> We also thank you for pointing out the confusing parts, we will add more descriptions in the main manuscript:
> - In fig2B and D, the model results were the average results across 12 models: MDS of the average RDM across 12 models for Figure 2B, and average RSA across 12 models for Figure 2D.
> - Decoding analysis details were stated in Appendix 4.2, and we will add more details in the manuscript. For each ROI or model layer, we trained linear decoders to classify the stimulus identity across 8 subclasses (Gifford et al., 2025) such as noise, natural scenes, manipulated scenes, contrast modulation, phase-coherence modulation, words, spiral gratings, chromatic noise.
> - For the statement that the decoders trained in encoding and generalized to the retrieval phase:
>   - It is expected that cross-phase decoding shows a lower accuracy compared to when it’s tested in the encoding phase, and this performance drop signals a large representational transformation (Xu 2025) in the early regions. We made the interpretation of a “better generalization” in the frontal regions due to their low performance drop. We will add description  in the manuscript.
>   - For both model and human results, when the decoders were tested in the retrieval phase, the decoding accuracy across layers were higher than the chance level 0.125 (permutation test $p < .01$), indicating a reliable generalization.
>
> **(Responses continued in the following comment)**

---

> > ### Author Response · Authors · 2025-12-04
> > **Rebuttal responses to Reviewer mji9 (3/3)**
> >
> > **8. The choice of 1-back dataset and the generalization**
> >
> > Our main research question – how WM representations are transformed from encoding to retrieval across the cortical and model layer hierarchy – could successfully fit in the paradigm of 1-back task. The 1-back task requires concurrent maintenance of stored targets and encoding of new ones, encouraging an active transformation of WM representation if there exist any. Several studies used only 1-back task to investigate the memory mechanisms combined with humans fMRI (Malisza et al 2005; Ricciardi et al 2006; Lee et al 2013; Ateş et al 2017) and EEG recordings (Audrain et al 2020; Gjini et al 2007).
> >
> > We appreciate the reference to Lei et al. (2024). While their work offers an impressive exploration of 9 tasks, it is a pure modeling study without any comparison with actual neural data. This allows them to generate infinite synthetic data across arbitrary task spaces. In contrast, **our study is grounded in the shared mechanisms between models and biological ground truth**. We are constrained by the availability of high-resolution, single-trial fMRI data on naturalistic stimuli, for which the NSD 1-back dataset is currently the state-of-the-art benchmark. Therefore, we focused on alignment with available biological data rather than broad task generalization. We acknowledge this in the Limitation in the revision.
> >
> > Additionally, to address the reviewer’s concern from model side, we have now added a 2-back task to investigate models’ representation transformation in response point 1 to Reviewer ppZL (**Supplementary Figure A4**). By comparing encoding and retrieval phase WM representations in the 2-back task, we observed a similar pattern of dynamic to stable WM coding across model layer hierarchy in both 1- and 2-back tasks.
> >
> > **9. Potential use of the state-space models**
> >
> > We appreciate the suggestion. In this work, we systematically evaluated 12 model configurations spanning 2 encoder architectures (CNN, ViT), 2 encoder learning objectives (supervised, self-supervised), and 3 recurrent modules (vanilla RNN, GRU, LSTM). These choices were motivated by their widespread use in neuroscience-inspired modeling and their well-documented dynamical properties. However, we don't think that our results strongly hinge on the specific model architecture choice: The main conclusion (larger WM transformation in earlier layers and more stable code in later layers) were supported with results averaged across 12 models (Figure 2-3 and Figure 4B).
> >
> > Modern state-space models (SSMs, such as S4, H3, Mamba) formalize neural computation as continuous or discrete dynamical systems, and several recent studies have begun applying SSMs to sequence processing and learning (Gu et al 2021; Vankadara et al., 2024), while little has been done to model the human neural process of memory or decision making processes. We agree that incorporating SSMs is a promising future direction, and we will update the Limitations to acknowledge this point.
> >
> > **10. Direct comparison of model representations and brain activation**
> >
> > To clarify, our study does not evaluate models based on the brain responses predictivity from model activations during the task. Rather, we ask whether models and humans share similar representational transformations across WM phases.
> >
> > We discussed the cross-system RSA in the response point 7 to Reviewer ts76. We observed low RSA scores in general. Therefore, human brains and models holding distinctive representational structures could still show similar mechanisms of transformation across working memory encoding to retrieval phases.

---

### Official Review · Reviewer_1p9M · 2025-11-04

**Soundness:** 3
**Presentation:** 3
**Contribution:** 3
**Rating:** 6
**Confidence:** 4

**Summary:**

In this submission, the authors employ representational similarity analysis to investigate how items are encoded in both artificial and biological visual working memory. They evaluate three competing hypotheses regarding the similarity of representations during encoding and retrieval phases. Their findings indicate a mixed coding mechanism, with representational stability across these phases increasing along the cortical hierarchy. To further support this, the authors assess decoding accuracy using mismatched encoding and retrieval representations, providing additional evidence for the mixed coding mechanism across cortical areas. Finally, they perform an ablation study on various learning objectives and architectural choices to determine which configurations yield neural networks most consistent with brain data.

**Strengths:**

Strengths:
- The fields of neuroscience and cognitive science have long tried to better understand how items are encoded in visual working memory with many competing theories proposed to support the process. The current submission uses tools from AI and data science to suggest that the particular encoding transformation in biological (and artificial) neural networks is a function of a hierarchical processing.
- I personally find the decoding evidence in Sec 4.2.2 to be quite a creative way to show the varied encoding schemes employed by neural networks to store items in the working memory.
- The paper is theoretically sound, and the reported statistical analyses are conducted with commendable rigor.

**Weaknesses:**

- The current submission presents compelling evidence regarding how different cortical areas represent items in visual working memory. However, I am concerned that the observed enhanced stability in higher visual representations—a central aspect of the proposed mixed coding hypothesis—may be influenced by a greater bias in recordings from these higher areas. To address this potential confound, I recommend demonstrating that decoders trained and tested on retrieval phase data achieve higher accuracy than those trained on encoder phase data.
- The analysis of how training objectives and architectural choices affect alignment with brain data is intriguing, especially as it contrasts with previous work suggesting that self-supervised learning (SSL) objectives yield more human-like visual representations [1]. I encourage the authors to expand this section by explicitly relating their findings to prior studies that claim SSL methods produce models more aligned with brain data than those trained with supervised learning objectives.

References:
1. Zhuang, C., Yan, S., Nayebi, A., Schrimpf, M., Frank, M. C., DiCarlo, J. J., & Yamins, D. L. (2021). Unsupervised neural network models of the ventral visual stream. Proceedings of the National Academy of Sciences, 118(3), e2014196118.

**Questions:**

NA. Please refer to my review above

---

> ### Author Response · Authors · 2025-12-04
> **Rebuttal responses to Reviewer 1p9M**
>
> We thank the reviewer for their insightful comments. We are encouraged by the recognition of the theoretical significance, methodological novelty, and rigour analyses of the work.
> To address the issue raised, we added results of decoders trained and tested in the retrieval phase, and added discussions on how training objectives influence alignment.
>
> **1. Stability in the frontal regions is not a result of a higher bias**
>
> We agree with the reviewer’s insightful suggestion. We now add these results in the **Supplementary Figure A1A**. We also took this opportunity to update our decoding accuracy results with training and testing procedures that balanced the number of images per class (more details in Appendix 4.2).
>
> The reviewer’s assumption that the decoders trained and tested on retrieval phase data should achieve higher accuracy than those trained on encoder phase and tested on retrieval would only be correct if that brain region adopts a dynamic coding strategy across WM phases. Indeed, in Figure A1A, we observed that in some early and mid level visual regions (V3, LO, IPS permutation $p < .05$, marginal for V2 and hV4 $p = .088$).
>
> Conversely, if a brain region uses a stable code, the decoders trained and tested on retrieval phase and decoders trained on encode and tested on retrieval phase should show similar performance. This matches our results in FEF and dlPFC (permutation $p > .05$).
>
> Moreover, the decoding accuracy of a more stable ROI also depends on the SNR of the particular phase the decoders were trained or tested. Due to a generally higher SNR during the encoding than the retrieval phase (Iamshchinina et al 2021), the decoders trained on retrieval phase may be limited by the SNR. This lower SNR effectively offsets the advantage of within-phase training, resulting in performance comparable to decoders trained on the (high-SNR) encoding phase. This might account for the similar decoding accuracy of the two decoders in some regions (V1, VO, OTS).
>
> Lastly, the decoders trained and tested on retrieval phase and decoders trained on encode and tested on retrieval phase both exhibited higher accuracy than chance level (permutation $p < .01$ across all ROIs including FEF and dlPFC, FDR corrected), indicating the meaningful category information during the encoding, retrieval phase in the frontal area, and a meaningful generalization from encoding to retrieval phase.
>
> **2. The effect of supervised and self-supervised learning**
>
> We agree that our findings can be related to prior reports about learning objectives, and we are happy to clarify how our results relate to the literature.
>
> Previous studies comparing SL and SSL models have primarily evaluated (1) their predictivity of brain responses to naturalistic images based on the model activation and (2) the emergence of category structure. Within these two domains, the literature reports mixed results:
> - Several studies have found advantages of certain SSL models. Zhuang et al. (2021) showed that recent SSL models achieve comparable or higher prediction accuracy for V4 and IT responses than SL models.
> - Other work, however, highlights similarities or even SL advantages: The modern SSL methods (Conwell et al 2024; Konkle & Alvarez 2022) exhibit comparable brain predictivity with SL models, while early SL models showed stronger IT predictivity (Khaligh-Razavi & Kriegeskorte 2014). Though SL and SSL (MoCo, SimCLR, etc) models behave surprisingly similarly on category and image level errors (Geirhos et al 2020), Chen et al (2020) reported SL advantage in classification accuracy than SSL models with matched model size.
>
> Our study addresses a different question from these previous works. **We do not evaluate models based on the brain responses predictivity. Rather, we ask whether models and humans exhibit shared representational transformations across WM phases.** This leads to a different alignment criterion in our work. Thus, it is not surprising that conclusions may diverge from some studies focused on brain predictivity or category geometry.
>
> When we consider the emergence of human-like representational transformation as a subset of the above topic, our conclusion aligns with studies showed comparable performance between SSL and SL, or even advantage of SL. With model scale controlled, SSL prioritizes invariance over categorical structure, which may distort the temporal transformations relevant for WM (Konkle & Alvarez 2022).
>
> Crucially, our main theoretical finding—that earlier layers show larger representational transformations while later layers show more stable codes—is consistent across all 12 models regardless of architectures or training objectives (Figures 2–3 RSA and decoding results).
>
> We will revise the discussion section to clarify how our findings relate to previous work, and why evaluating dynamic WM transformations leads to different but complementary insights.

---

### Author Response · Authors · 2025-12-04
**Rebuttal response summary**

We sincerely thank all reviewers and the AC for their time, constructive feedback, and engagement, which have led to substantial improvements to our paper. During the rebuttal phase, we:

**1. Strengthen the key human evidence**

- We added a control area (A1) in both RSA and decoding analysis, as well as additional decoders both trained and tested in the WM retrieval phase. These results jointly support the active involvement and relatively stable coding strategy of frontal regions during WM processing.
- We conducted additional permutation tests for the human fMRI data (n=8). With stronger statistical reliability, the results consistently support that earlier visual areas exhibit larger representational transformations, whereas later areas maintain more stable codes.

**2. Extended the theoretical and model-based generalization**

- We added Limitation to acknowledge that the NSD 1-back dataset is currently the only available fMRI WM benchmark with naturalistic stimuli, and since our goal is to compare mechanisms of humans and models under matched conditions, our design is constrained by the availability of such biological ground truth.
- To address this on the model side, we analyzed models with fine-tuned encoders trained on the 1-back task and additionally trained new models on a 2-back task. The results converged to the same central conclusion: hierarchical representational transformations remain consistent regardless of task difficulty or whether encoders are frozen or fine-tuned.
- We also incorporated an additional non-linear RBF CKA analysis, which operates in a high-dimensional feature space and captures more complex representational relationships. This analysis provided convergent support for our main findings.

**3. Further clarified the theoretical framing and methodology**

- We added a paragraph in the Related Work section emphasizing that rather than assessing models by brain predictivity, we ask whether models exhibit human-like hierarchical patterns of representational transformations from WM encoding to retrieval. We also extended the Discussion section regarding the influence of training objectives.
- We clarified the conceptual novelty of our work in the first Discussion section by emphasizing hierarchical changes from dynamic to stable WM coding strategies and convergent evidence across humans and models.
- We improved methodological descriptions, including clearer presentation of error-bar computation, justification for the choice of top-1 and top-2 axis rotation angles, and rationale for the selected model architectures.

---

### Meta-Review · Area_Chair_Xij5 · 2026-01-06

**Summary:**

The reviewers raised three main concerns that drove the discussion of the decision. First, regarding scope and contribution, multiple reviewers (mji9, ppZL, and ts76) argued that the claims are too broad for a single visual 1-back benchmark. While the rebuttal acknowledges this limitation and adds 2-back results on the model side, generality on the human side remains largely untested. Second, regarding the interpretation of frontal "stability," reviewers [mji9, 1p9M] expressed concern that the dlPFC/FEF effects could be attributed to SNR/weak stimulus modulation rather than stable working memory (WM) coding. The rebuttal addresses this concern by introducing a control ROI, permutation tests, and additional retrieval-phase decoding. However, it does not fully settle the SNR/ROI-involvement alternative, and it also does not directly resolve mji9’s conceptual concern that an n-back task likely requires strongly dynamic coding (the rebuttal mainly reframes the claim as “relatively more stable / less transformed” rather than fully stable). Third, regarding robustness and methodology, reviewers asked for clearer uncertainty estimates and questioned the rotation/linearity assumptions. The rebuttal improves reporting and adds permutation testing and a nonlinear CKA analysis.

There were additional concerns, such as architecture breadth, novelty framing, and training-objective interpretation. Assessing whether all of the changes in the rebuttal fully resolve these concerns would require a careful re-review of the revised manuscript and new supplementary results. Although I assume some reviewers would have raised their score, the paper is still just marginally below the acceptance threshold.

**Reviewer Concerns:**

**Concerns weakly resolved or outstanding**

Task scope / generality beyond NSD 1-back [mji9, ppZL, ts76]
- limitation is acknowledged and 2-back is added on the model side
- human evidence is still from one task/dataset, so breadth of contribution remains a concern

Conceptual concern that an n-back task likely requires a strongly dynamic code [mji9]
- no direct answer, instead a reframing as “relatively more stable / less transformed” coding in frontal ROIs rather than a fully stable representation

Frontal ROI interpretation (SNR / ROI involvement) [mji9]
- control ROI helps, but there is still no direct ROI-wise SNR/reliability analysis

Training objective interpretation (SL vs SSL) [1p9M, ts76]
- rebuttal clarifies that their alignment target differs from brain-predictivity
- a direct test of what drives SL vs SSL differences is still missing

Modeling breadth (attention/SSM, broader training distributions) [mji9, ts76]
- mainly acknowledged as future work, not tested in this revision

**Addressed or largely addressed:**

Frontal ROI interpretation (stable code vs analysis artifact) [1p9M, mji9]
- the rebuttal adds a control ROI (A1) and shows chance-level decoding there
- adds retrieval→retrieval decoding and compares it to encode→retrieval
- these additions strengthen the claim that frontal ROIs carry task-relevant information

Statistics / reporting [mji9, ts76]
- now switched to permutation tests (important given n=8)
- error bar computation is explained
- multi-seed model results are added

Metrics / rotation analysis choices [mji9, ppZL, ts76]
- clearer rationale for using top-1/top-2 axes in the rotation analysis
- adds a nonlinear RBF CKA analysis as a convergent check

Frozen encoder concern [ppZL, ts76]
- adds fine-tuned encoder model results and reports similar main conclusions

**Reviewer Scores:**

Reviewer 1p9M (initial: 6): remains at 6 or might have increased, not so likely. Although most concerns were addressed.
Reviewer mji9 (initial: 2): would increase to 4. The rebuttal addressed some of his concerns. The reviewer was not wrong in any of his statements, so I count his concerns into the decision.
 Reviewer ppZL (initial: 4): might have increased to 6. Most concerns were answered (2-back models, fine-tuned encoders, and a nonlinear metric (CKA).
Reviewer ts76 (initial: 4): remains at 4, also skeptical on novelty.

---

### Decision · Program_Chairs · 2026-01-26

Reject